# An armoured marine reptile from the Early Triassic of South China and its phylogenetic and evolutionary implications

Andrzej S Wolniewicz[1,2], Yuefeng Shen[1], Qiang Li[1,3], Yuanyuan Sun[4], Yu Qiao[1], Yajie Chen[1], Yi-Wei Hu[1], Jun Liu[1]*

[1]School of Resources and Environmental Engineering, Hefei University of Technology, Hefei, China; [2]Institute of Paleobiology, Polish Academy of Sciences, Warsaw, Poland; [3]Section Paleontology, Institute of Geosciences, University of Bonn, Bonn, Germany; [4]Chengdu Center, China Geological Survey (Southwest China Innovation Center for Geosciences), Chengdu, China

*For correspondence:
junliu@hfut.edu.cn

**Competing interest:** The authors declare that no competing interests exist.

**Abstract** Sauropterygia was a taxonomically and ecomorphologically diverse clade of Mesozoic marine reptiles spanning the Early Triassic to the Late Cretaceous. Sauropterygians are traditionally divided into two groups representing two markedly different body plans – the short-necked, durophagous Placodontia and the long-necked Eosauropterygia – whereas Saurosphargidae, a small clade of armoured marine reptiles, is generally considered as the sauropterygian sister-group. However, the early evolutionary history of sauropterygians and their phylogenetic relationships with other groups within Diapsida are still incompletely understood. Here, we report a new saurosphargid from the Early Triassic (Olenekian) of South China – *Prosaurosphargis yingzishanensis* gen. et sp. nov. – representing the earliest known occurrence of the clade. An updated phylogenetic analysis focussing on the interrelationships among diapsid reptiles recovers saurosphargids as nested within sauropterygians, forming a clade with eosauropterygians to the exclusion of placodonts. Furthermore, a clade comprising *Eusaurosphargis* and *Palatodonta* is recovered as the sauropterygian sister-group within Sauropterygomorpha tax. nov. The phylogenetic position of several Early and Middle Triassic sauropterygians of previously uncertain phylogenetic affinity, such as *Atopodentatus*, *Hanosaurus*, *Majiashanosaurus*, and *Corosaurus*, is also clarified, elucidating the early evolutionary assembly of the sauropterygian body plan. Finally, our phylogenetic analysis supports the placement of Testudines and Archosauromorpha within Archelosauria, a result strongly corroborated by molecular data, but only recently recovered in a phylogenetic analysis using a morphology-only dataset. Our study provides evidence for the rapid diversification of sauropterygians in the aftermath of the Permo-Triassic mass extinction event and emphasises the importance of broad taxonomic sampling in reconstructing phylogenetic relationships among extinct taxa.

## Editor's evaluation

This significant new study is built around a remarkable fossil of a new genus and species of armoured marine reptile from the Early Triassic of China. More importantly, this paper also adds to our understanding of the diversification of reptiles during the Triassic and also sheds light on the interrelationships of a wide range of important groups. The authors present a solid phylogenetic analysis of sauropterygians, which reveals a possible new clade, Sauropterygomorpha, which is recovered close to Archosauromorpha. The authors suggest that the latter, as well as the Testudines and Ichthyosauromorpha, belong to a clade that had previously only been recovered using phylogenomic data, the

Archelosauria. This paper will be of interest to a broad range of scientists, including palaeontologists, herpetolgists, and evolutionary biologists.

## Introduction

Several groups of reptiles invaded the marine realm in the aftermath of the Permo-Triassic mass extinction (PTME), the largest extinction event in Earth's history (*Benton, 2015*). This phenomenon was likely a result of the scarcity of marine competitors and predators caused by the PTME and high productivity in the incipient shallow marine environment (*Vermeij and Motani, 2018*). Triassic marine reptiles, including the iconic Ichthyosauromorpha and Sauropterygia, as well as some other smaller and lesser known groups, achieved high taxonomic and ecological diversity rapidly after their emergence in the late Early Triassic and played a pivotal role in the reorganisation of marine food webs following the PTME (*Scheyer et al., 2014*; *Kelley and Pyenson, 2015*; *Motani et al., 2015*; *Motani et al., 2017*; *Jiang et al., 2016*; *Stubbs and Benton, 2016*; *Cheng et al., 2019*; *Moon and Stubbs, 2020*; *Li and Liu, 2020*; *Sander et al., 2021*; *Qiao et al., 2022*). Because Mesozoic marine reptiles represent likely several independent transitions from a terrestrial to an aquatic lifestyle, they also provide an ideal system to analyse the roles of function and constraint in determining evolutionary pathways (*Motani, 2009*; *Benson, 2013*).

Sauropterygia was a diverse clade of Mesozoic marine reptiles that first appeared in the late Early Triassic (*Jiang et al., 2014*; *Li and Liu, 2020*) and its members remained important predators in marine ecosystems until their extinction at the end of the Late Cretaceous (*Bardet, 1994*; *Rieppel, 2000a*; *Benson and Druckenmiller, 2014*). Sauropterygia is traditionally divided into two major lineages, representing two markedly different body plans – the Placodontia and the Eosauropterygia, the latter comprising Pachypleurosauria, Nothosauroidea and Pistosauroidea (which includes the iconic Plesiosauria; *Rieppel, 2000a*). Placodonts were characterised by the presence of short necks and short skulls and possessed crushing palatal dentition almost certainly used for feeding on hard-shelled invertebrates (*Rieppel, 2002*; *Crofts et al., 2017*). Some early-diverging placodonts had a limited covering of osteoderms on their backs and possibly limbs (*Jiang et al., 2008*; *Klein and Scheyer, 2013*), but derived forms evolved extensive dorsal armour superficially similar to that of turtles (*Scheyer, 2007*; *Wang et al., 2020*). Eosauropterygians, on the other hand, had elongated necks and elongated skulls with pointed dentition suitable for capturing fast-moving prey (*Rieppel, 2000a*; *Rieppel, 2002*). Placodonts remained restricted to shallow marine environments until their extinction in the Late Triassic, whereas eosauropterygians evolved a suite of adaptations for a pelagic lifestyle and became one of the dominant groups of marine reptiles in the Jurassic and Cretaceous (*Kelley et al., 2014*).

Even though sauropterygians have a rich fossil record and a long history of scientific research, the early evolution of the group is still incompletely understood. In a broad phylogenetic context, sauropterygians have been consistently recovered within Diapsida, but their exact phylogenetic position relative to other diapsid groups remains unresolved. Several diapsid clades, including Saurosphargidae (*Li et al., 2014*; *Li et al., 2018*; *Neenan et al., 2015*; *Wang et al., 2022*), Testudines (*deBraga and Rieppel, 1997*; *Schoch and Sues, 2015*), Ichthyosauromorpha (*Martínez et al., 2021*), Thalattosauria (*Simões et al., 2022*) and the armoured reptile *Eusaurosphargis dalsassoi* (*Scheyer et al., 2017*) were previously proposed to be the sauropterygian sister-group, but its exact identity is still a matter of debate. Because sauropterygians were suggested as being closely related not only to turtles, but also to archosauromorphs (*Chen et al., 2014a*; *Neenan et al., 2015*; *Martínez et al., 2021*; *Simões et al., 2022*), resolving their phylogenetic placement within diapsids is of crucial importance for solving the phylogenetic uncertainty surrounding Archelosauria – a clade comprising turtles and archosauromorphs strongly supported by molecular data (*Crawford et al., 2015*; *Lyson and Bever, 2020*), but only recently recovered in a phylogenetic analysis using a morphology-only dataset (*Simões et al., 2022*).

Saurosphargids are a small clade of Mesozoic marine reptiles characterised by the presence of body armour comprising broadened dorsal ribs, forming a closed 'rib-basket', and a moderately- to well-developed osteoderm covering (*Li et al., 2011*; *Li et al., 2014*). Saurosphargids are known from the Middle Triassic of Europe and South China, although recent evidence suggests they could have survived as late as the Late Triassic (*Scheyer et al., 2022*). Saurosphargids comprise as many as four

**eLife digest** Around 252 million years ago, just before the start of a period of time known as the Triassic, over 90% of animals, plants and other species on Earth went extinct in what was the worst mass extinction event in the planet's history. It is thought to have happened because of an increase in volcanic eruptions that led to global warming, acid rain and other catastrophic changes in the environment.

The loss of so many species caused ecosystems to restructure as the surviving species evolved to fill niches left by those that had gone extinct. On land, reptiles diversified to give rise to dinosaurs, the flying pterosaurs, and the ancestors of modern crocodiles, lizards, snakes and turtles. Some of these land-based animals evolved to live in water, resulting in many species of marine reptiles emerging during the Triassic period.

This included the saurosphargids, a group of marine reptiles that lived in the Middle Triassic around 247–237 million years ago. They were 'armoured' with a shield made of broadened ribs superficially similar to that of turtles, and a covering of bony plates. However, it is unclear how the saurosphargids evolved and how closely they are related to other marine reptiles.

Here, Wolniewicz et al. studied a new species of saurosphargid named *Prosaurosphargis yingzishanensis* that was found fossilized in a quarry in South China. The animal was around 1.5 metres long and had a chest shield and armoured plates like other saurosphargids. The characteristics of the rock surrounding the fossil suggest that this individual lived in the Early Triassic, several million years before other saurosphargid species.

The team used a phylogenetic approach to infer the evolutionary relationships between *P. yingzishanensis* and numerous other land-based and marine reptiles based on over 220 anatomical characteristics of the animals. The resulting evolutionary tree indicated that the saurosphargids represented an early stage in the evolution of a larger group of marine reptiles known as the sauropterygians. The analysis also identified the closest land-based relatives of sauropterygians.

These findings provide evidence that marine reptiles rapidly diversified in the aftermath of the mass extinction event 252 million years ago. Furthermore, they contribute to our understanding of how ecosystems recover after a major environmental crisis.

taxa – *Saurosphargis volzi* (considered a nomen dubium by some authors) (*Huene, 1936*; *Nosotti and Rieppel, 2003*; *Scheyer et al., 2017*), the heavily armoured *Sinosaurosphargis yunguiensis* (*Li et al., 2011*; *Hirasawa et al., 2013*), and two species in the genus *Largocephalosaurus* (*L. polycarpon* and *L. qianensis*; *Cheng et al., 2012*; *Li et al., 2014*). Saurosphargids are one of the reptile groups proposed as the sister-group of sauropterygians (see above), but some recent phylogenetic analyses have suggested their placement within sauropterygians instead, as either the sister-group to placodonts (*Schoch and Sues, 2015*; *Simões et al., 2018*; *Martínez et al., 2021*) or eosauropterygians (*Scheyer et al., 2017*; *Wang et al., 2019*; *Simões et al., 2022*).

Our understanding of the early evolution of the sauropterygian body plan is also hindered by the uncertain phylogenetic position of several Early and Middle Triassic taxa. *Hanosaurus hupehensis* (*Young, 1972*; *Rieppel, 1998a*; *Wang et al., 2022*) and *Majiashanosaurus discocoracoidis* (*Jiang et al., 2014*) from the Early Triassic of South China are variably recovered as either lying outside the clade comprising Saurosphargidae + Sauropterygia (*Hanosaurus*; *Wang et al., 2022*), as pachypleurosaurs (*Jiang et al., 2014*; *Neenan et al., 2015*; *Lin et al., 2021*; *Wang et al., 2022*), or as outgroups to a clade comprising nothosauroids and pachypleurosaurs to the exclusion of pistosauroids (*Li and Liu, 2020*). The phylogenetic placement of *Corosaurus alcovensis* from the Early–Middle Triassic of Wyoming, USA, is also unresolved, with different authors arguing for its early-diverging eosauropterygian (*Rieppel, 1994*; *Li and Liu, 2020*), eusauropterygian (more closely related to nothosauroids and pistosauroids than to pachypleurosaurs; *Neenan et al., 2015*; *Lin et al., 2021*), or pistosauroid (*Rieppel, 1998b*; *Wang et al., 2022*) affinity. The phylogenetic position of the herbivorous hammer-headed sauropterygian *Atopodentatus unicus* from the Middle Triassic of South China relative to placodonts and eosauropterygians also remains uncertain (*Cheng et al., 2014*; *Li et al., 2016*; *Wang et al., 2022*). These conflicting phylogenetic placements are likely the result of inadequate sampling of Early and Middle Triassic sauropterygian ingroup taxa, as well as the inclusion of only a limited

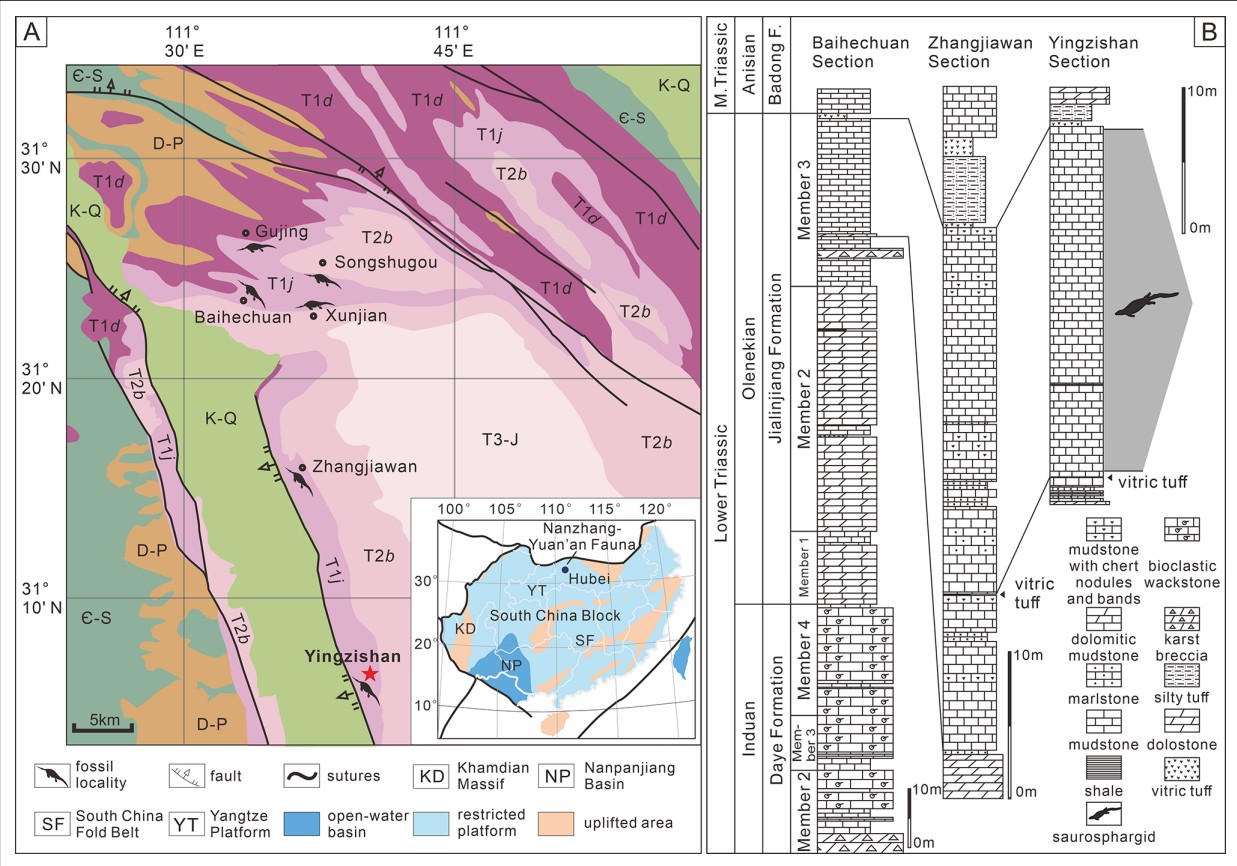

**Figure 1.** Locality and horizon of *Prosaurosphargis yingzishanensis* (HFUT YZSB-19-109). (**A**) The geological map of the Nanzhang-Yuan'an region showing Yingzishan quarry, where HFUT YZSB-19-109 was collected; inset is a paleogeographic map of the South China Block in the Triassic showing the location of the Nanzhang-Yuan'an fauna (after **Qiao et al., 2019**). (**B**) Stratigraphic column showing the horizon from which HFUT YZSB-19-109 was collected. Abbreviations: Є-S, Cambrian–Silurian; D-P, Devonian–Permian; F., Formation; K-Q, Cretaceous–Quaternary; M., Middle; T1*d*, Daye Formation, Lower Triassic; T1*j*, Jialingjiang Formation, Lower Triassic; T2*b*, Badong Formation, Middle Triassic; T3-J, Upper Triassic–Jurassic.

number of diapsid reptile clades, including potential sauropterygian outgroups, in previous phylogenetic analyses.

Here, we report an Early Triassic saurosphargid from South China, representing the earliest recorded occurence of the group. We also present an updated phylogenetic hypothesis for Diapsida with a particular focus on Sauropterygia, which we use as context for discussing the early evolutionary assembly of the sauropterygian body plan.

## Results
### Geological background

Yingzishan quarry, where the new specimen (HFUT YZSB-19-109) was collected, is located on the northern boundary of the Yangtze Platform (*Figure 1A*; *Li and Liu, 2020*). The new specimen represents the Early Triassic Nanzhang-Yuan'an fauna (*Li and Liu, 2020*) and originates from the upper part of the third member of the Jialingjiang Formation (*Figure 1B*). Traditionally, the Jialingjiang Formation has been divided into four members by most authors studying the region (*Li et al., 2002*; *Zhao et al., 2008*; *Cheng et al., 2015*). From base to top, the first member consists of thick-bedded to massive dolostone, the second member consists of vermicular limestone intercalated with dolostone, the third member consists of laminated thin- to medium-bedded limestone to dolostone, whereas the fourth member is composed of dolostone and karstified breccia. Thus, in the four-member division of the Jialingjiang Formation, the thick volcanic ash (*Figure 1B*) marks the bottom of the fourth member. However, recent geological mapping in the region (*Chen et al., 2016a*) proposed that the fourth

member of the Jialingjiang Formation should be included in the Middle Triassic Badong Formation, indicating a three-member division of the Jialingjiang Formation. This division is consistent with the lithology of the Jianlingjiang and Badong formations as defined by the official geological guide of Hubei Province (*Bureau of Geology and Mineral Resources of Hubei Province, 1990*). The division of the Lower and Middle Triassic in the region is also consistent with the widespread thick volcanic ash as a marker of the Lower–Middle Triassic boundary in the Yangtze Platform (*Chen et al., 2020*). Consequently, this three-member division was followed by *Cheng et al., 2019*, *Li and Liu, 2020*, *Qiao et al., 2019* and is also followed in this study.

However, the second and third members of the Jialingjiang Formation were recently redefined by *Yan et al., 2021*, who regarded the lowermost base of the thick vitric tuff as the boundary between the second and third members (contra *Chen et al., 2016a*; *Cheng et al., 2019*). As a consequence, the Nanzhang-Yuan'an fauna was placed into the newly defined second member of the Jialingjiang Formation (*Cheng et al., 2022*; *Zhao et al., 2022*). We argue that this new definition and division of different members of the Jialingjiang Formation contradicts the official geological guide (*Bureau of Geology and Mineral Resources of Hubei Province, 1990*) and causes confusion. Therefore, we prefer to maintain the definition of the thick vitric tuff as the boundary between the Early Triassic Jialingjiang and the Middle Triassic Badong formations in the Nanzhang-Yuan'an region (*Figure 1B*), pending future updates of the official geological guide of Hubei Province.

The new specimen is buried in dark grey, laminated, and thin-bedded carbonate mudstone (*Figure 1B*) with some carbonaceous interactions. There are also some peloids, replacive dolomites, and microbial mats in the fossiliferous levels (*Chen et al., 2022*). Based on the published sedimentological accounts (*Wang et al., 2011*; *Chen et al., 2016b*; *Yan et al., 2018*; *Yan et al., 2021*; *Li et al., 2020*; *Zhao et al., 2022*) and field observations, a restricted, stagnant, and hypersaline lagoon within a tidal flat environment is inferred as the burial setting of the marine reptiles of the Nanzhang-Yuan'an fauna (*Chen et al., 2022*).

## Systematic palaeontology

Reptilia *Laurenti, 1768*
Diapsida *Osborn, 1903*
Archelosauria *Crawford et al., 2015*
Sauropterygomorpha tax. nov.

Definition: The most recent common ancestor of *Eusaurosphargis dalsassoi* and *Pistosaurus longaevus*, and all of its descendants (min ∇ *Eusaurosphargis dalsassoi* Nosotti and Rieppel, 2003 & *Pistosaurus longaevus Meyer, 1839*).

Diagnosis: Osteoderms present (ch. 143.1), body strongly flattenned dorso-ventrally (ch. 144.1), clavicle applied to the medial surface of scapula (ch. 148.1), metatarsal V long and slender (ch. 186.0), metatarsal I less than 50% the length of metatarsal IV (ch. 189.1).

Sauropterygia *Owen, 1860*
Saurosphargidae *Li et al., 2011*
*Prosaurosphargis yingzishanensis* gen. et sp. nov.
urn:lsid:zoobank.org:act:36BED757-A86D-4951-B664-A91969F7CDBF       urn:lsid:zoobank.org:act:825E86DD-E636-4A99-AA85-7AF4DBCD7E8B

Holotype: HFUT YZSB-19-109, a partial postcranial skeleton (*Figure 2*). The specimen is housed in the Geological Museum of Hefei University of Technology, Hefei, Anhui Province, China (HFUT).

Etymology: Genus name from the Greek preposition π ρ ό (pró), meaning before, and *Saurosphargis*, the name of the type genus of the family Saurosphargidae (*Li et al., 2011*). The specific epithet refers to the type locality (see above).

Horizon and locality: Third Member of the Jialingjiang Formation (uppermost Spathian, Olenekian, Lower Triassic), Yingzishan quarry, Yuan'an County, Hubei Province, China.

Diagnosis: A saurosphargid characterised by the following combination of character states: (1) spaces between dorsal transverse processes anteroposteriorly shorter than the anteroposterior widths of the transverse processes (like in *Saurosphargis* and *Sinosaurosphargis*, but different from *Largocephalosaurus*, in which the spaces between the dorsal transverse processes are wider than

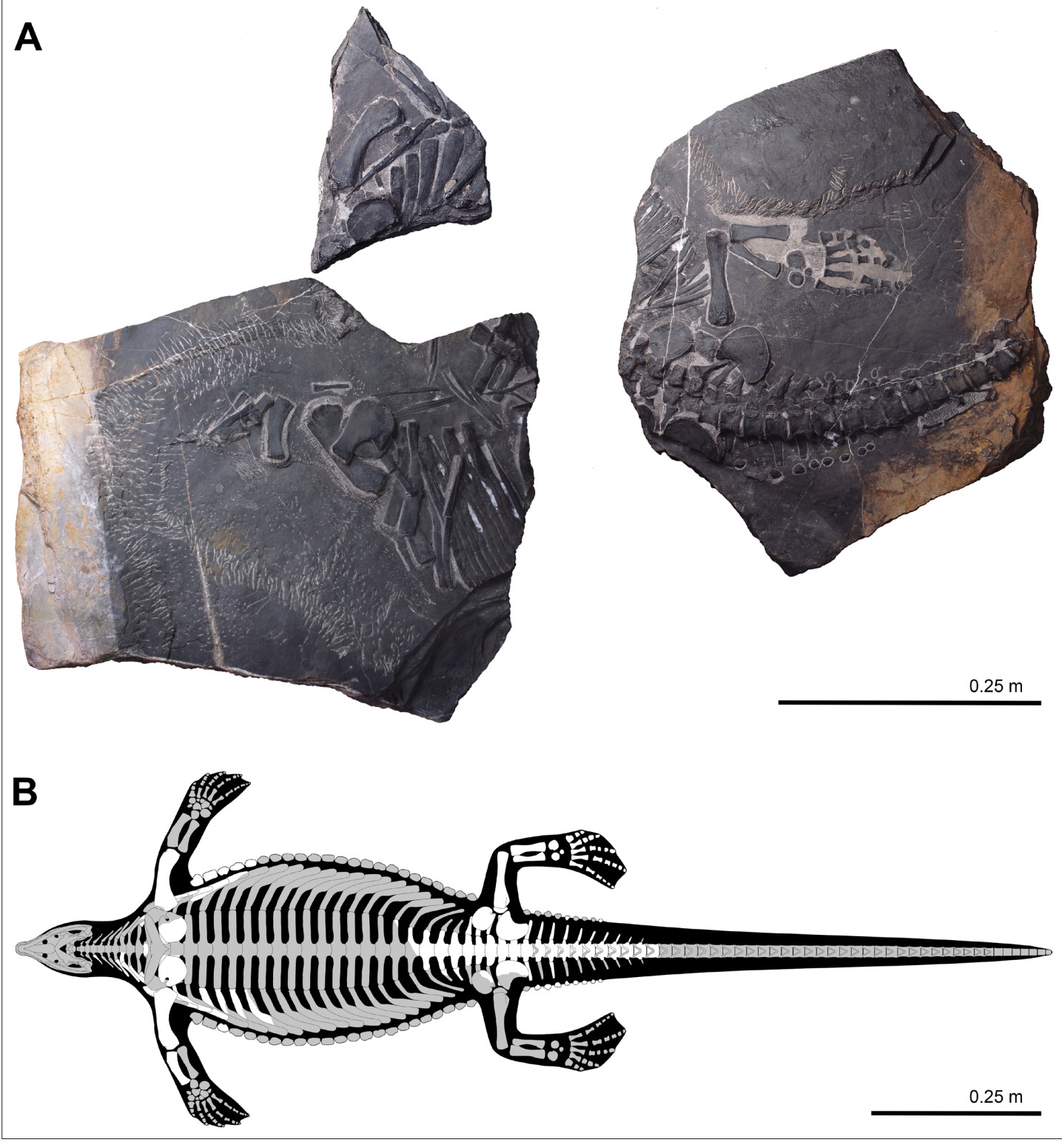

**Figure 2.** Holotype of *Prosaurosphargis yingzishanensis* (HFUT YZSB-19-109) (**A**) and skeletal reconstruction with known elements highlighted in white (gastralia removed for clarity) (**B**).

the transverse processes); (2) ribs without uncinate processes (like in *Sinosaurosphargis*, but unlike in *Saurosphargis* and *Largocephalosaurus*, in which uncinate processes are present); (3) osteoderms forming a median, parasaggital and lateral rows (similar to *Largocephalosaurus polycarpon*, but different from *L. qianensis*, in which additional, small osteoderms more extensively cover the lateral sides of the body, and different from *Sinosaurosphargis*, in which the osteoderms form an extensive dorsal armour); (4) ectepicondylar groove on humerus present (like in *Largocephalosaurus qianensis*, absent in *L. polycarpon*); (5) entepicondylar foramen in humerus absent (as in *Largocephalosaurus*) (details of humerus morphology unknown in *Sinosaurosphargis*); (6) radius short relative to humerus compared with other saurosphargids; (7) presence of a single distal tarsal (distal tarsal IV) (different from *Largocephalosaurus*, which possesses two distal tarsals – III and IV) (number of tarsals unknown in *Sinosaurosphargis*); (8) anterior caudal ribs shorter than sacral ribs (anterior caudal ribs longer than sacral ribs in *Largocephalosaurus*, unknown in *Sinosaurosphargis*).

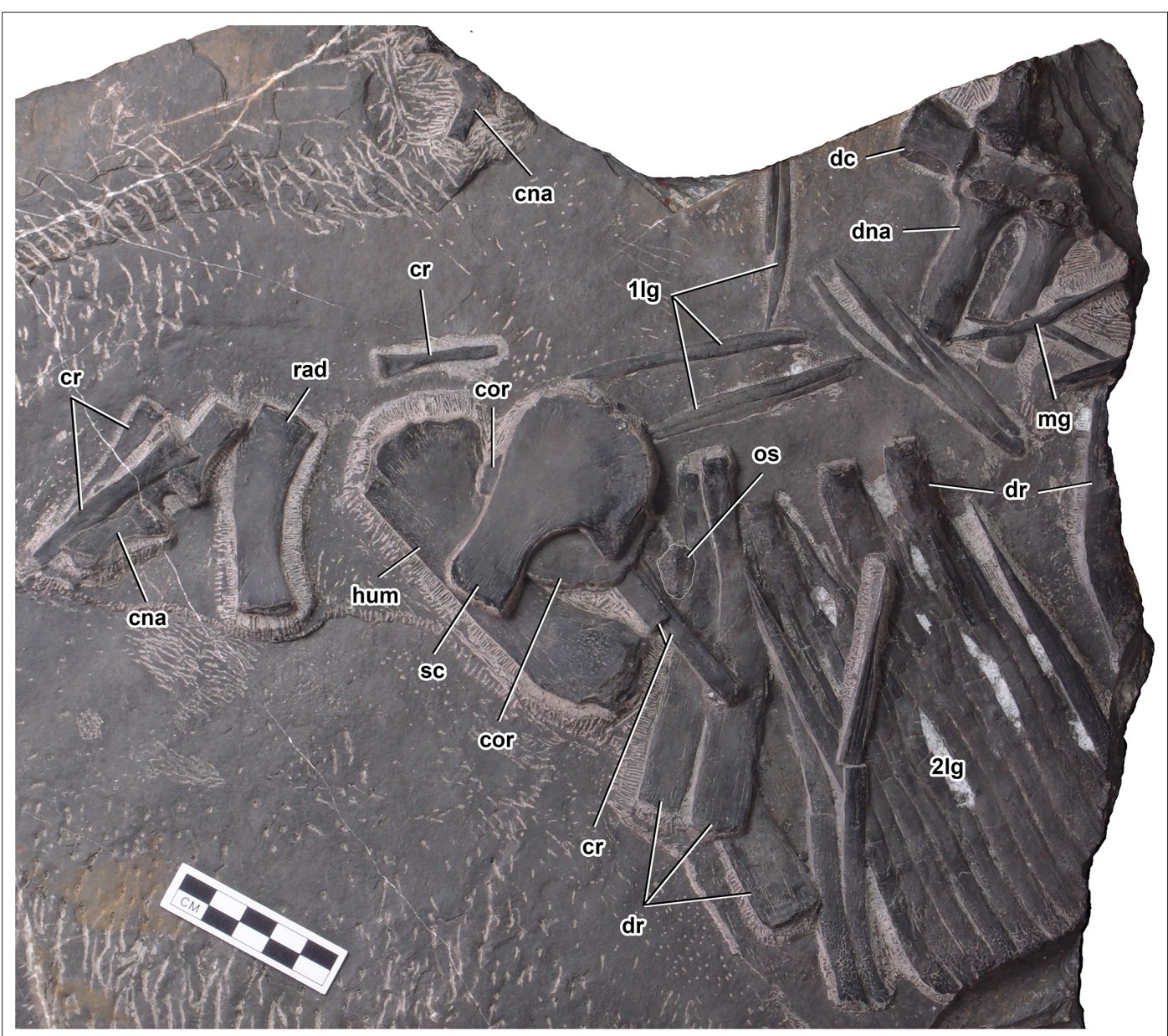

**Figure 3.** Anterior vertebral column, right pectoral girdle, and right forelimb elements of *Prosaurosphargis yingzishanensis*. Abbreviations: 1lg, first lateral gastral element; 2lg, second lateral gastral element; cna, cervical neural arch; cor, coracoid; cr, cervical rib; dc, dorsal centrum; dna, dorsal neural arch; dr, dorsal rib; hum, humerus; mg, median gastral element; os, osteoderm; rad, radius; sc, scapula. Scale bar = 5 cm.

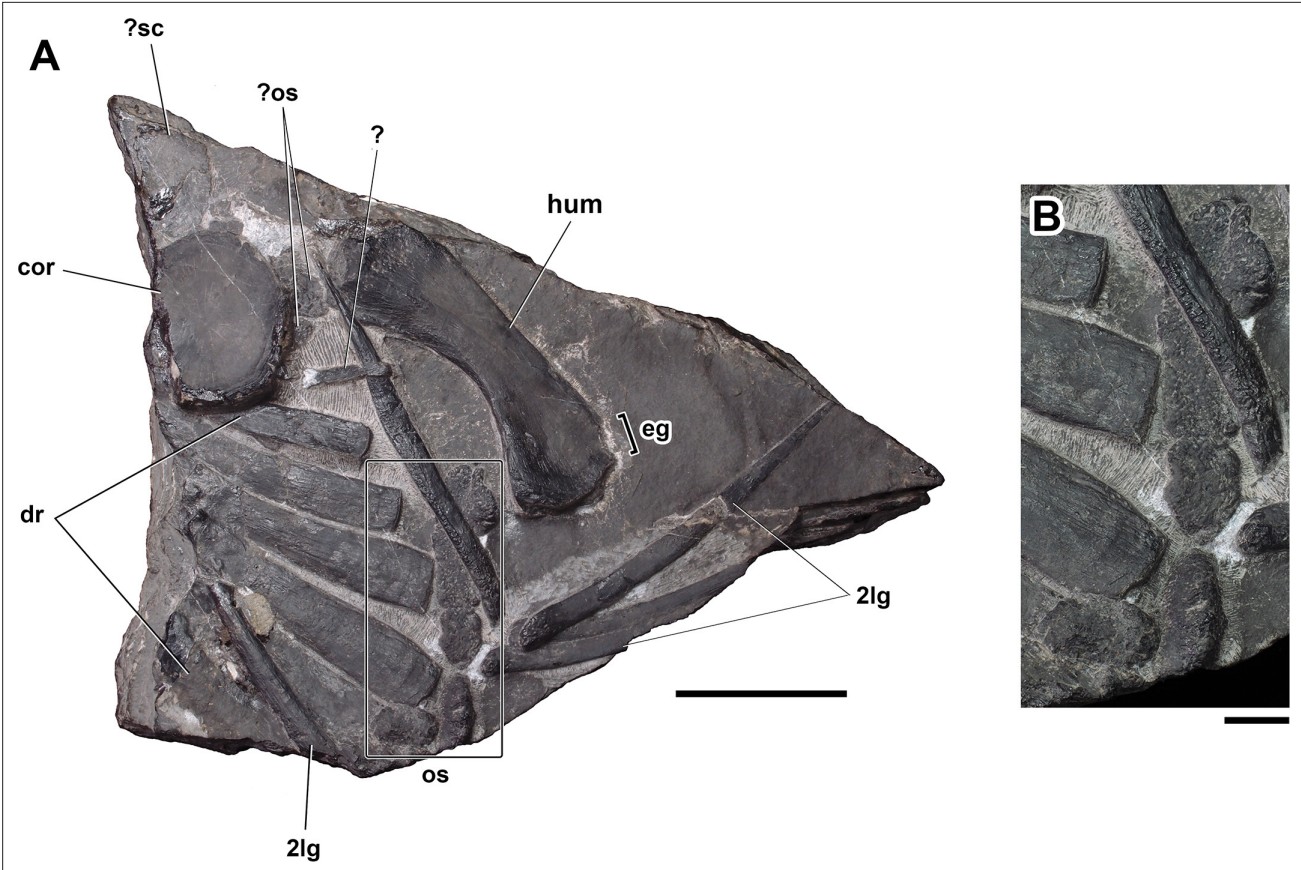

**Figure 4.** Dorsal ribs, osteoderms, coracoid, and humerus from the left side of the body of *Prosaurosphargis yingzishanensis* (**A**) and detail of lateral osteoderms (**B**). Abbreviations: 2lg, second lateral gastral element; cor, coracoid; dr, dorsal rib; eg, ectepicondylar groove; hum, humerus; os, osteoderm, ?sc, ?scapula. Scale bar = 5 cm in (**A**) and 1 cm in (**B**).

## Description and comparisons

HFUT YZSB-19-109 comprises three blocks (*Figure 2*). The first large block contains mostly disarticulated parts from the anterior right portion of the postcranial skeleton – 2 cervical neural arches and 5 cervical ribs, 1 dorsal centrum and 2 dorsal neural arches (all three preserved in articulation), 9 dorsal ribs, 1 median, 11 lateral and 13 lateralmost gastral elements, a single parasaggital osteoderm, the right scapula, right coracoid, right humerus, and right radius (*Figure 3*). The second small block contains the distal ends of the six most anterior dorsal ribs, five lateralmost gastral elements, as many as eight osteoderms, a partial left coracoid, and a left humerus (*Figure 4*). The third large block preserves the articulated posterior part of the body, including 5 posterior dorsal neural arches and associated ribs, 4 posterior dorsal/sacral vertebrae and ribs, an articulated series of 11 anterior caudals with associated ribs, chevrons and osteoderms, 14 lateralmost gastralia, a single parasaggital osteoderm, a partial right and a complete left pelvic girdle, and a complete left hindlimb (*Figures 5 and 6*). Based on the humerus:total body length and femur:total body length ratios of the type specimen of *Largocephalosaurus qianensis* (specimen IVPP V 15638, total body length = 2317 mm; *Li et al., 2014*), the only completely preserved saurosphargid specimen discovered to date, the total length of HFUT YZSB-19-109 is estimated to have reached between 1468 mm to 1583 mm, respectively. Selected measurements of HFUT YZSB-19-109 are given in *Table 1*.

### Axial skeleton

Vertebrae: The vertebral column of HFUT YZSB-19-109 is represented by two disarticulated cervical neural arches, a single anterior dorsal centrum and two anterior dorsal neural arches (all three preserved in articulation), and an articulated series of 20 vertebrae comprising five posterior dorsal neural arches, four complete posterior dorsal/sacral vertebrae and 11 complete caudal vertebrae.

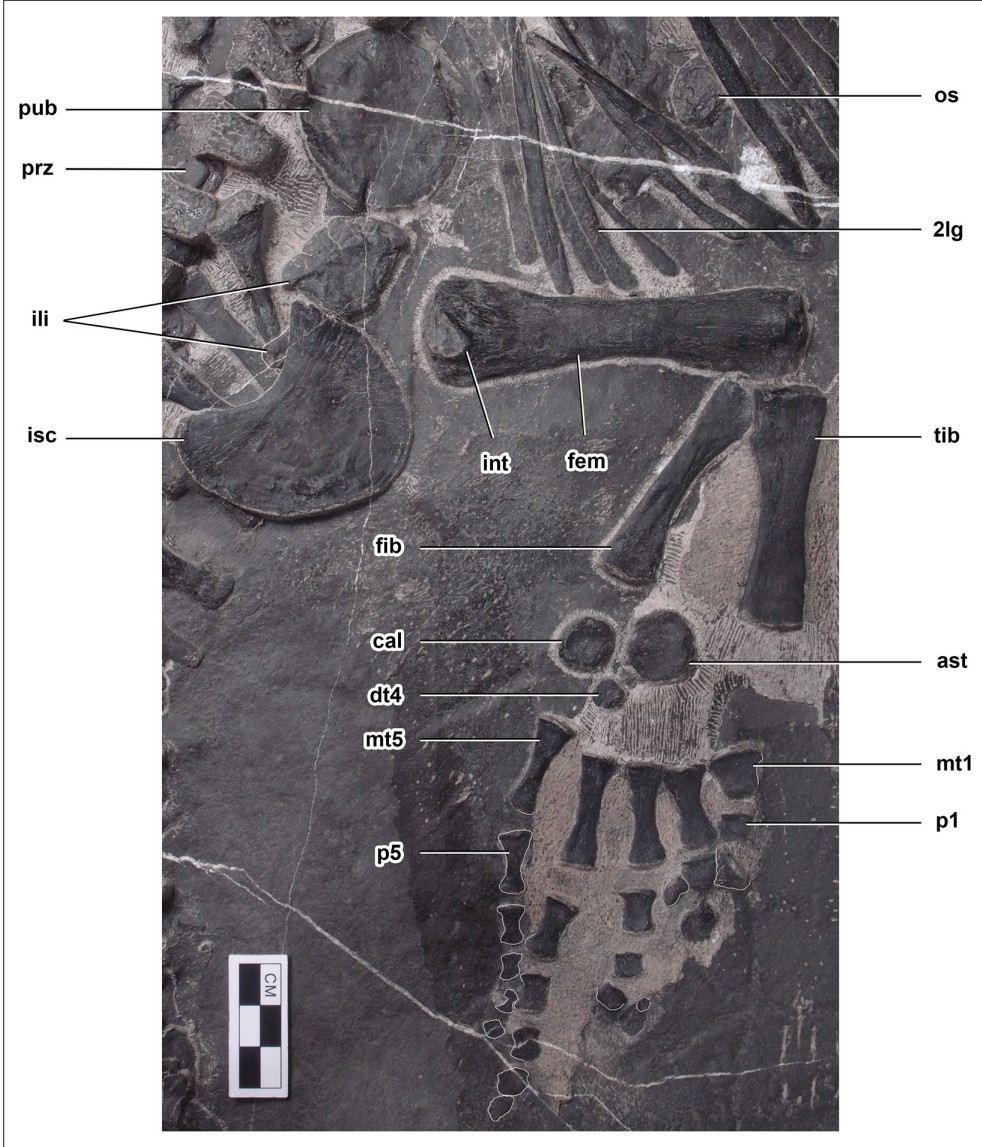

**Figure 5.** Pelvic girdle and hindlimb of *Prosaurosphargis yingzishanensis*. Abbreviations: 2lg, second lateral gastral element; ast, astragalus; cal, calcaneum; dt4, distal tarsal IV; fem, femur; fib, fibula; ili, ilium; int, internal trochanter; isc, ischium; mt1, metatarsal I; mt5, metatarsal V; os, osteoderm; p1, first phalanx of digit 1; p5, first phalanx of digit 5; pub, pubis; prz, prezygapophysis; tib, tibia. Scale bar = 3 cm.

Two cervical neural arches are preserved in HFUT YZSB-19-109 (*Figure 3*). Only the left side of a small neural arch, representing one of the anterior post-axial cervicals, is preserved in dorsal view, whereas a larger neural arch, belonging to one of the posterior cervicals, is completely preserved and exposed in ventral view, but it is partially obscured by an overlying cervical rib and is rotated by approximately 180 degrees, so that its anterior end faces posteriorly. The transverse processes of the cervical neural arches extend posterolaterally and are proportionally shorter than the transverse processes of the dorsal neural arches. A well-developed prezygapophysis is clearly visible in the smaller neural arch. Well-developed postzygapophyses are preserved in both the anterior and posterior cervical neural arches, where they are visible protruding from just underneath the overlying cervical rib in the latter. Approximately oval, anteroposteriorly elongate facets for articulation with the centrum are exposed in the posterior cervical neural arch.

The only preserved dorsal centrum is exposed in ventral view (*Figure 3*). It is mediolaterally constricted at its anteroposterior mid-length, resembling the dorsal centra of other saurosphargids (*Huene, 1936*; *Li et al., 2011*; *Li et al., 2014*). The pedicels of the dorsal neural arches are

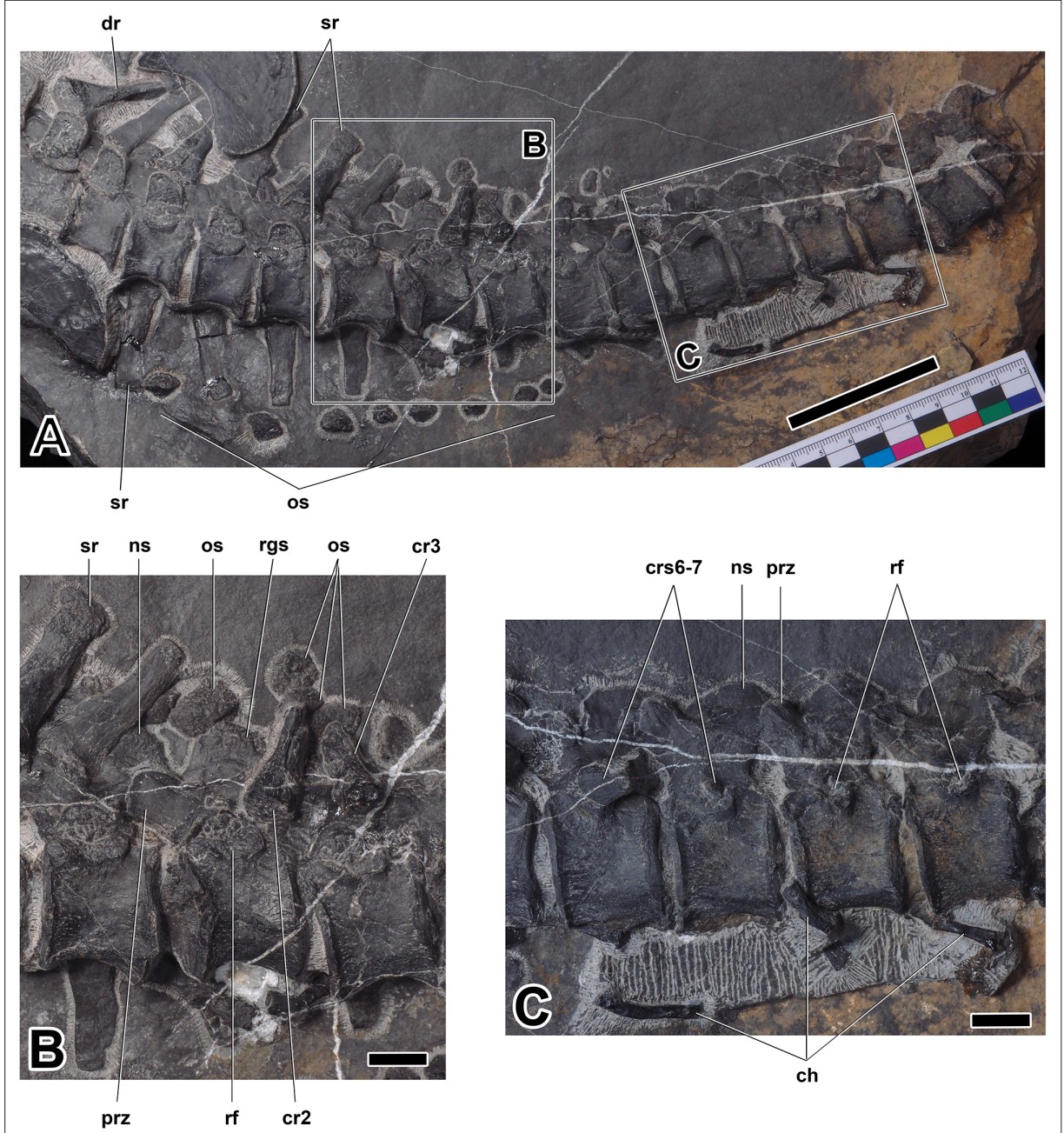

**Figure 6.** Sacral and caudal vertebrae of *Prosaurosphargis yingzishanensis* (**A**), close-up of anterior caudal vertebrae (**B**), and close-up of more posterior caudal vertebrae (**C**). Abbreviations: ch, chevron; cr2, second caudal rib; cr3, third caudal rib; crs6-7, sixth and seventh caudal ribs; dr, dorsal rib; ns, neural spine; os, osteoderm; prz, prezygapophysis; rf, rib facet; rgs, rugose surface; sr, sacral rib. Scale bar = 5 cm in (**A**) and 1 cm in (**B**) and (**C**).

anteroposteriorly elongate and mediolaterally narrow, resembling those preserved in *Saurosphargis* (*Huene, 1936*). The transverse processes are straight and mediolaterally elongate. The maximum width of the transverse process is greater than the maximum width of the space between the transverse processes of adjacent neural arches (*Figure 3*), a condition similar to that exhibited by *Saurosphargis* (*Huene, 1936*) and *Sinosaurosphargis* (*Li et al., 2011*), but different from *Largocephalosaurus* (*Li et al., 2014*) and *Eusaurosphargis* (*Scheyer et al., 2017*), in which the widths of the spaces between the dorsal transverse processes are approximately equal or greater than the widths of the transverse processes.

**Table 1.** Selected measurements of HFUT YZSB-19-109, holotype of *Prosaurosphargis yingzishanensis*.

| Vertebral column | |
|---|---|
| **Centrum position** | **Maximum anteroposterior length (measured along mid-dorsoventral height)** |
| Last dorsal | 18.25 mm |
| 1st sacral | 19.45 mm |
| 2nd sacral | 20.49 mm |
| 3rd sacral | 20.50 mm |
| 1st caudal | 20.50 mm |
| 2nd caudal | 18.35 mm |
| 3rd caudal | 19.40 mm |
| 4th caudal | 20.14 mm |
| 5th caudal | 18.64 mm |
| 6th caudal | 19.10 mm |
| 7th caudal | 17.95 mm |
| 8th caudal | 18.84 mm |
| 9th caudal | 18.10 mm |
| 10th caudal | 17.86 mm |
| **Pectoral girdle** | |
| Right scapula | |
| Maximum proximodistal length | 68.09 mm |
| Maximum proximal width | 51.20 mm |
| Right coracoid | |
| Maximum proximodistal length | 51.65 mm |
| Left coracoid | |
| Maximum proximodistal length | 52.25 mm |
| Maximum anteroposterior length | 39.40 mm |
| **Forelimb** | |
| Right humerus | |
| Maximum proximodistal length | 104.44 mm |
| Maximum proximal width | 38.85 mm |
| Maximum distal width | 27.80 mm |
| Left humerus | |
| Maximum anteroposterior length | 104.24 mm |
| Minimum mediolateral width | 20.10 mm |
| Maximum distal width | 31.50 mm |
| Right radius | |
| Maximum proximodistal length (measured along proximodistal axis) | 58.50 mm |
| Maximum proximodistal length (measured along anterior margin) | 60.95 mm |

*Table 1 continued on next page*

*Table 1 continued*

| Vertebral column | |
|---|---|
| **Centrum position** | **Maximum anteroposterior length<br>(measured along mid-dorsoventral height)** |
| Maximum proximodistal length<br>(measured along posterior margin) | 51.10 mm |
| **Pelvic girdle** | |
| Left ilium | |
| Maximum proximodistal length | 23.20 mm |
| Maximum distal width | 32.99 mm |
| Left ischium | |
| Maximum anteroposterior (proximodistal) length | 51.44 mm |
| Maximum mediolateral width | 36.59 mm |
| Left pubis | |
| Maximum proximodistal length | 43.25 mm |
| Maximum anteroposterior length | 36.85 mm |
| Hindlimb | |
| Left femur | |
| Maximum proximodistal length | 93.85 mm |
| Maximum proximal width | 27.65 mm |
| Maximum distal width | 23.40 mm |
| Minimum mediolateral (anteroposterior) width | 14.25 mm |
| Left tibia | |
| Maximum proximodistal length | 60.35 mm |
| Maximum proximal width | 18.29 mm |
| Maximum distal width | 16.39 mm |
| Minimum mediolateral (anteroposterior) width | 10.90 mm |
| Left fibula | |
| Maximum proximodistal length | 55.59 mm |
| Maximum proximal width | 12.90 mm |
| Maximum distal width | 15.24 mm |
| Minimum mediolateral (anteroposterior) width | 7.25 mm |
| Astragalus | |
| Maximum proximodistal length | 18.45 mm |
| Maximum mediolateral (anteroposterior) width | 18.20 mm |
| Calcaneum | |
| Maximum proximodistal length | 14.25 mm |
| Maximum mediolateral (anteroposterior) width | 13.55 mm |
| Distal tarsal IV | |
| Maximum proximodistal length | 9.49 mm |
| Maximum mediolateral (anteroposterior) width | 8.95 mm |
| Metacarpals | |

*Table 1 continued on next page*

*Table 1 continued*

| | Vertebral column | |
|---|---|---|
| **Centrum position** | **Maximum anteroposterior length (measured along mid-dorsoventral height)** | |
| Metacarpal I maximum proximodistal length | 13.44 mm | |
| Metacarpal II maximum proximodistal length | 20.89 mm | |
| Metacarpal III maximum proximodistal length | 25.89 mm | |
| Metacarpal IV maximum proximodistal length | 28.19 mm | |
| Metacarpal V maximum proximodistal length | 24.25 mm | |

Five posterior dorsal neural arches are preserved in articulation in HFUT YZSB-19-109 and are exposed in ventral view. In adddition, a fragmentary centrum is associated with the second preserved posterior dorsal neural arch. The articulation surfaces for the centra located on the pedicels of the posterior dorsal neural arches are anteroposteriorly elongate, but seem mediolaterally broader and not as well demarcated from the transverse processes as those in the more anterior dorsal neural arches. The posterior dorsal transverse processes gradually become mediolaterally shorter and antero-posteriorly broader posteriorly. Well-developed prezygapophysis-postzygapophysis articulations are preserved and exposed between the second and fifth posterior dorsal neural arches. Articulated anterior and posterior dorsal neural arches preserved without their respective centra indicate a lack of fusion of the neurocentral suture, a paedomorphic feature characteristic for Sauropterygia (*Rieppel, 2000a*) and also present in *Saurosphargis* (*Huene, 1936*) and *Largocephalosaurus* (*Li et al., 2014*).

Four complete posterior dorsal/sacral vertebrae are preserved and exposed in left ventrolateral view. Two sacral vertebrae can be identified by the presence of associated sacral ribs with clearly expanded distal ends. However, the distal ends of the first two ribs lying immediately posterior to the last unambiguous dorsal rib are obscured by the overlying ilium and ischium, so it is not possible to confidently determine whether they represent the last two dorsal ribs or the first two sacral ribs (*Figure 6A*). A series of 11 articulated and complete caudal vertebrae is also preserved, with two anterior caudals exposed in left ventrolateral view and the remaining caudals exposed in lateral view (*Figure 6*). The sacral and caudal centra have concave lateral and ventral surfaces and are likely amphicoelus, as evidenced by the concave anterior articular surface of the second caudal centrum, but in most cases the articular surfaces are obscured by matrix or adjacent centra, giving a false impression of procoely or amphiplaty. The caudal centra are dorsoventrally taller than anteroposteriorly long and gradually decrease in size posteriorly. The caudal neural arches bear well-developed, anterodorsally inclined prezygapophyses. The neural spines of some caudal neural arches are exposed in lateral view (*Figure 6B*). They are dorsoventrally short, anteroposteriorly broad and possess a convex dorsal margin. A slightly rugose/striated dorsal surface, reminiscent of a similar surface present in the dorsal neural spines of *Helveticosaurus* (*Rieppel, 1989a*), *Augustasaurus* (*Sander et al., 1997*), *Nothosaurus* (*Klein et al., 2022*), and *Pomolispondylus* (*Cheng et al., 2022*), is preserved in some of the anterior caudal neural spines. Rib facets are visible from the second sacral to the penultimate preserved caudal vertebra (*Figure 6*). The dorsal portions of the rib facets are located on the ventrolateral surfaces of the neural arches, whereas their ventral portions are located on the dorsolateral surfaces of the centra and are demarcated by a prominent, ventrally arcuate ridge (*Figure 6B*). The surface of the rib facets is rugose. The last visible minute caudal rib is preserved in articulation with the seventh caudal vertebra, but small rib facets are also preserved in the following three vertebrae (*Figure 6C*). It is not clear, however, if small ossified caudal ribs were associated with these vertebrae in life.

Ribs: Six cervical ribs are preserved in HFUT YZSB-19-109 (*Figures 3 and 4*). Three anterior cervical ribs are approximately straight and possess a single, expanded head and narrow distal ends. Two posterior cervical ribs are also preserved, but their proximal portions are obscured by the overlying coracoid and scapula. These ribs are much longer than the anterior cervical ribs and possess slightly curved shafts and relatively broad distal ends. HFUT YZSB-19-109 also preserves 16 dorsal ribs, but most of them are incomplete and/or obscured by overlying skeletal elements, such as gastralia or other ribs (*Figures 3, 4 and 6A*). The anterior dorsal ribs are much more robust than the posterior

cervical ribs, being proximodistally longer and anteroposteriorly much broader (*Figures 3 and 4*). They have a single head and greatly expanded distal portions that abut against each other, forming a characteristic 'rib-basket' also present in other saurosphargids (*Huene, 1936*; *Li et al., 2011*; *Li et al., 2014*), but differ from the dorsal ribs of *Eusaurosphargis*, which are narrow and widely spaced (*Scheyer et al., 2017*). The dorsal ribs do not bear a distinct uncinate process, being similar in this respect to the dorsal ribs of *Sinosaurosphargis* (*Li et al., 2011*), but differ from the dorsal ribs of *Largocephalosaurus*, *Saurosphargis*, *Eusaurosphargis* and a possible isolated saurosphargid rib from the Late Triassic of Switzerland, all of which possess uncinate processes (*Li et al., 2014*; *Scheyer et al., 2017*; *Scheyer et al., 2022*).

Five to seven posterior dorsal ribs are preserved in HFUT YZSB-19-109; they have an expanded head, a narrow distal end and are much shorter and slender in comparison with the more anterior dorsal ribs. As a consequence, they did not form part of the closed 'rib-basket'. Two to four pairs of sacral ribs are preserved in HFUT YZSB-19-109, although the right sacral ribs are damaged and partially preserved as impressions (see above regarding the uncertainty in establishing the correct number of sacral vertebrae/rib pairs; *Figure 6A*). Two unambiguous sacral ribs are proximodistally slightly longer than the last evident dorsal rib and possess clearly expanded distal ends. The penultimate left sacral rib bears a small posteroproximal process. The caudal ribs are proximodistally short, possess a broad head and taper distally (*Figure 6*). They are not fused with the caudal centra. The anterior caudal ribs in HFUT YZSB-19-109 are shorter than the sacral ribs, in contrast to *Largocephalosaurus*, in which the anterior caudal ribs are longer than the sacral ribs (*Li et al., 2014*). The caudal ribs decrease in size posteriorly and extend at least to the level of the seventh caudal centrum, although it is not clear if caudal ribs extended posteriorly beyond the seventh caudal (see above). In *Largocephalosaurus*, the caudal ribs extend to the level of the 10th or 11th caudal centrum (*Li et al., 2014*).

Chevrons: Eight chevrons are preserved in HFUT YZSB-19-109 in association with some of the posterior caudal vertebrae (*Figure 6A and C*). In lateral view, the chevrons are slender and straight or display a gentle ventral curvature. The chevrons have a proximal end that is approximately equal in size or only slightly expanded relative to the distal end.

Gastralia: The gastral rib basket is partially preserved in HFUT YZSB-19-109 (*Figures 3–5*). Like in *Eusaurosphargis*, *Placodus* and eosauropterygians (*Drevermann, 1933*; *Rieppel, 1989b*; *Nosotti and Rieppel, 2003*; *Shang et al., 2011*), each gastral rib was composed of five elements – a median element, two lateral (first lateral) elements and two lateralmost (second lateral) elements. Gastral ribs comprising five elements were also inferred for *Saurosphargis* (*Huene, 1936*), whereas in *Sinosaurosphargis* the gastral ribs were described as comprising three elements – one median and two lateral elements (*Li et al., 2011*), and the number of gastral rib elements was not specified for *Largocephalosaurus* (*Li et al., 2014*). A single anterior median gastral element, 11 lateral and 33 lateralmost elements are preserved in HFUT YZSB-19-109. The median element is weakly angulated anteriorly, whereas the first lateral elements form proportionally short rods with a blunt medial end and a pointed lateral end (*Figure 3*) and closely resemble the first lateral gastral elements of *Placodus* (*Drevermann, 1933*). The lateralmost elements are concentrated into two articulated series comprising 13 (anterior block; *Figure 3*) and 14 (posterior block; *Figure 5*) elements each. They form mediolaterally elongate rods which are approximately straight or gently angulated posteriorly. The medial end of the second lateral element is narrow and tapers into a pointed apex, whereas the lateral end is broadened and blunt with a distinctly rugose/granulated surface. One anteriorly positioned first lateral element seems to bifurcate laterally, forming two lateral prongs (*Figure 3*).

## Osteoderms

Several osteoderms are preserved in HFUT YZSB-19-109. A single osteoderm is preserved in the anterior block between the anterior dorsal ribs (*Figure 3*) and a second isolated osteoderm is preserved in the posterior block between the posterior gastral elements (*Figure 5*). In addition, two possible osteoderms are also preserved in association with the proximal end of the left humerus (*Figure 4*). These osteoderms are approximately oval in outline and likely represent osteoderms forming parasagittal rows similar to those in *Largocephalosaurus* (*Li et al., 2014*; *Scheyer et al., 2022*). A partial series comprising six osteoderms is preserved along the distal ends of a series of partially preserved left dorsal ribs (*Figure 4*). These osteoderms represent the left lateral osteoderm row and are anteroposteriorly elongate, have an irregular, sub-oval, or sub-rectangular outline, and a densely pitted surface.

One of the posterior elements in this series bears a prominent ridge extending along the midline of its exposed surface. In all these features, these osteoderms closely resemble the lateral osteoderms reported for *Saurosphargis* (*Huene, 1936*) and *Largocephalosaurus polycarpon* (*Li et al., 2014*).

Another partial series of small osteoderms is preserved lateral to the distal ends of the right sacral and anterior caudal ribs, indicating that the lateral osteoderm rows extended to the level of the seventh caudal vertebra (*Figure 6A*). A few small osteoderms are also preserved in close association with the anterior caudal neural spines and seem to have formed a median row along the dorsal midline (*Figure 6B*), likely overlying the caudal neural spines in a manner similar to that in *Largocephalosaurus* (*Li et al., 2014*) and *Placodus* (*Jiang et al., 2008*). In *L. qianensis*, a dense covering of small osteoderms is present on the dorsal surface of the neck, trunk and caudal region (*Li et al., 2014*), but no such osteoderms were found in association with HFUT YSZB-19-109. A dense covering of osteoderms is also absent in *L. polycarpon* (*Cheng et al., 2012*; *Li et al., 2014*).

## Pectoral girdle

Scapula: The right scapula is completely preserved in HFUT YZSB-19-109 (*Figure 3*). In addition, a large and broad broken bone fragment likely representing the left scapula is visible lying anterodorsally to the left coracoid (*Figure 4*). The scapula possesses an anteroproximally expanded glenoid portion and a much narrower, straight, and posterodorsally projecting scapular blade. The glenoid portion forms a low acromion process proximodorsally. The coracoid facet is relatively broad and convex, whereas the glenoid facet is straight and short and oriented nearly parallel to the long axis of the scapular blade. The scapular blade is separated from the glenoid portion by a deep posteroventral notch. The scapular blade is straight, with gently concave anterodorsal and posteroventral margins and an approximately straight posterior margin. In general shape and proportions, the scapula of

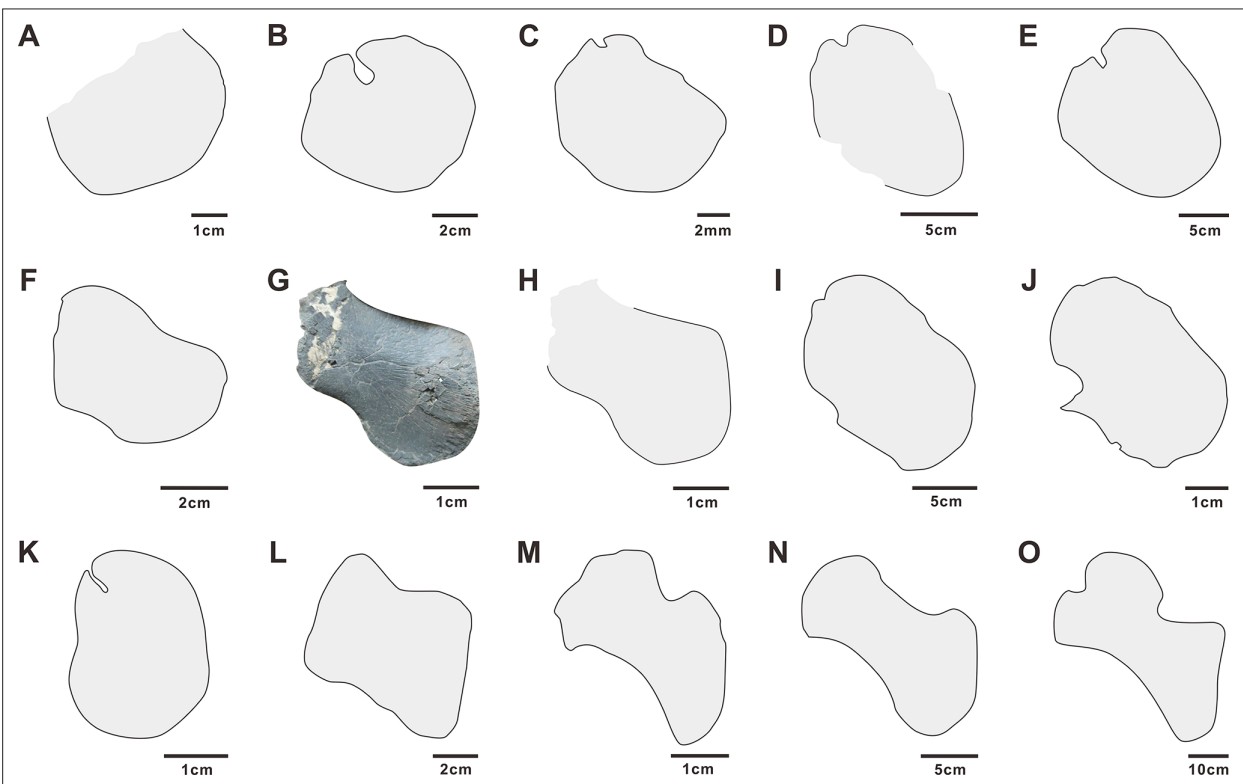

**Figure 7.** Coracoids of selected Early and Middle Triassic sauropterygians. A - *Prosaurosphargis* (based on specimen HFUT YZSB-19-109); B - *Largocephalosaurus* (after *Li et al., 2014*); C - *Eusaurosphargis* (after *Scheyer et al., 2017*); D - *Atopodentatus* (after *Cheng et al., 2014*); E - *Placodus* (after *Drevermann, 1933*); F - *Paraplacodus* (after *Rieppel, 2000b*); G (photo) and H (outline) - *Hanosaurus* (holotype, based on specimen IVPP V 3231); I - *Hanosaurus* (referred specimen; after *Wang et al., 2022*); J - *Lariosaurus sanxiaensis* (after *Li and Liu, 2020*); K – *Majiashanosaurus* (after *Jiang et al., 2014*), L – *Corosaurus* (after *Storrs, 1991*); M - *Wumengosaurus* (after *Wu et al., 2011*); N – *Anarosaurus* (after *Klein, 2012*); O – *Nothosaurus* (after *Ji et al., 2014*).

HFUT YZSB-19-109 closely resembles the scapula of *Largocephalosaurus* (*Cheng et al., 2012*; *Li et al., 2014*) and *Corosaurus* (*Storrs, 1991*), although the notch separating the glenoid portion from the scapular blade is much deeper and narrower in the latter.

Coracoid: The coracoids are only partially visible in HFUT YZSB-19-109 – the right coracoid is almost entirely covered by the right scapula (*Figure 3*), whereas the proximal portion of the left coracoid is broken (*Figure 4*). The preserved parts of both coracoids indicate that it was a dorsoventrally flat, plate-like element, approximately sub-circular in outline, closely resembling the coracoid of *Largocephalosaurs* (*Li et al., 2014*; *Figure 7*). The coracoid of HFUT YZSB-19-109 differs, however, from the coracoids of *Helveticosaurus* (Rieppel 1989), *Eusaurosphargis*, *Atopodentatus*, early diverging placodonts, *Majiashanosaurus*, a referred specimen of *Hanosaurus*, and *Lariosaurus sanxiaensis*, in which the coracoid is proximodistally more elongate and approximately sub-oval in outline (*Figure 7*). It also differs from the coracoids of eosauropterygians, which possess weakly- or well-developed anterior and posterior emarginations (*Figure 7*). A small notch in the proximal part of the right coracoid, seen just above the anterior margin of the right scapula, likely represents the coracoid foramen. The exposed ventral surface of the left coracoid bears numerous radial striations extending from the centre of the bone towards its outer margins.

## Forelimb

Humerus: Both humeri are preserved in HFUT YZSB-19-109 – the right humerus is complete, but partially overlapped by the right scapular blade (*Figure 3*), whereas the left humerus is slightly broken proximally (*Figure 4*). The humerus is proximodistally elongate; it is also posteriorly curved along its proximodistal axis, like the humerus of other saurosphargids (*Li et al., 2011*; *Li et al., 2014*; *Cheng et al., 2012*), *Placodus* (*Jiang et al., 2008*), *Majiashanosaurus* (*Jiang et al., 2014*), *Lariosaurus sanxiaensis* (*Li and Liu, 2020*), a referred specimen of *Hanosaurus* (*Wang et al., 2022*) and numerous eosauropterygians (*Rieppel, 1994*). However, in contrast to *Largocephalosaurus*, the anterior margin of the humerus is not convex, but straight, making it more similar to the humeri of *Placodus* and *Nothosaurus* (*Rieppel, 1994*). The shape of the anterior margin of the humerus is unknown in *Sinosaurosphargis*. The posterior margin of the humerus is concave. The proximal and distal ends of the humerus are expanded, but a distinct humeral head and distal condyles are not present. Anteriorly, the distal end of the humerus bears a shallow ectepicondylar groove (notch), like in *Largocephalosaurus qianensis* (*Li et al., 2014*), but unlike in *L. polycarpon* (*Cheng et al., 2012*), which possesses an entepicondylar foramen. Anteroproximally, the exposed surface of the right humerus bears a short, straight groove, which likely represents a muscle insertion site.

Radius: A disarticulated right radius is completely preserved in HFUT YZSB-19-109 in close proximity to the right humerus and right pectoral girdle (*Figure 3*). The radius is proximodistally elongate and anteriorly curved, with a convex anterior margin and a concave posterior margin. The proximal and distal ends of the radius are straight and slightly expanded, with the proximal end being anteroposteriorly broader than the distal end. The radius is similar in general shape to that in other saurosphargids (*Li et al., 2011*; *Li et al., 2014*; *Cheng et al., 2012*), *Majiashanosaurus* (*Jiang et al., 2014*), *Placodus* (*Jiang et al., 2008*), and *Helveticosaurus* (*Rieppel, 1989a*), but differs from the straight and anteriorly and posteriorly concave radius of *Eusaurosphargis* (*Scheyer et al., 2017*). The radius/humerus proximodistal length ratio in HFUT YZSB-19-109 is ~0.55, in line with *Eusaurosphargis* (~0.51–0.57; *Scheyer et al., 2017*) and *Placodus* (~0.56; *Jiang et al., 2008*), but is significantly smaller than the ratio in *Sinosaurosphargis* (~0.66; *Li et al., 2011*), *Largocephalosaurus polycarpon* (~0.74; *Cheng et al., 2012*), *L. qianensis* (~0.67–0.72; *Li et al., 2014*), a referred specimen of *Hanosaurus* (~0.67; *Wang et al., 2022*), and *Majiashanosaurus* (~0.74; *Jiang et al., 2014*). This indicates that HFUT YZSB-19-109 had a proportionally shorter forearm compared with other saurosphargids and early eosauropterygians.

## Pelvic girdle

Ilium: The left ilium is completely preserved in HFUT YZSB-19-109 and is exposed in lateral or medial view (*Figure 5*). The ilium consists of a distally expanded acetabular portion and a posteriorly projecting iliac blade, which is largely obscured by the overlying ischium. The ilium of HFUT YZSB-19-109 is very similar to the ilia of *Largocephalosaurus* (*Li et al., 2014*), *Eusaurosphargis* (*Nosotti and*

*Rieppel, 2003*), *Placodus* (*Rieppel, 1995*), and *Corosaurus* (*Storrs, 1991*), which all possess a well-developed, posteriorly projecting iliac blade.

Pubis: The left pubis is completely preserved in HFUT YZSB-19-109 and exposed in ventral view, whereas only the distal portion of the right pubis is preserved (*Figure 5*). The pubis is approximately oval in outline, being proximodistally longer than anteroposteriorly broad and bears a posteroproximally positioned, open obturator foramen. The pubis of HFUT YZSB-19-109 resembles the pubis of *Largocephalosaurus* (*Li et al., 2014*) in outline and proportions. It is also similar to the pubis of the holotype and referred specimens of *Hanosaurus* (*Rieppel, 1998a*; *Wang et al., 2022*) and *Pararcus diepenbroekii* (*Klein and Scheyer, 2013*), which are however more circular in outline, being approximately as wide proximodistally as long anteroposteriorly. The pubis of HFUT YZSB-19-109 differs markedly from the anteriorly and posteriorly shallowly emarginated pubis of *Eusaurosphargis* (*Scheyer et al., 2017*) and *Placodus* (*Drevermann, 1933*), and the deeply emarginated pubis of eosauropterygians (e.g. *Rieppel, 2000a*).

Ischium: The left ischium of HFUT YZSB-19-109 is completely preserved, but is rotated 180° relative to its life position and is exposed in dorsal view, whereas only the distal portion of the right ischium is preserved (*Figure 5*). The ischium forms an anterodistally convex and posteroproximally concave plate, similar to that in *Largocephalosaurus* (*Li et al., 2014*), *Pararcus* (*Klein and Scheyer, 2013*) and *Hanosaurus* (*Rieppel, 1998a*; *Wang et al., 2022*), but differs markedly from the anteriorly emarginated ischium of *Eusaurosphargis* (*Scheyer et al., 2017*) and the anteriorly and posteriorly emarginated ischia of some placodonts and eosauropterygians (*Rieppel, 2000a*).

## Hindlimb

Femur: The left hindlimb is completely preserved in HFUT YZSB-19-109. The femur is exposed in posterior view (*Figure 5*). The shaft of the femur is straight and both the proximal and distal ends are expanded. The internal trochanter is well-developed and located proximally, as in *Largocephalosaurus* (*Li et al., 2014*), *Simosaurus*, and *Nothosaurus* (*Rieppel, 1994*), but differs from the more distally located trochanter in the holotype of *Hanosaurus* (*Rieppel, 1998a*). Distally, the femur produces weakly-developed, but still distinct, condyles for the tibia and fibula, separated ventrally by a shallow popliteal area. The humerus/femur ratio in HFUT YZSB-19-109 is ~1.19, which is similar to the ratio in *Largocephalosaurus* (~1.20; *Li et al., 2014*), but is significantly greater than the ratio in *Helveticosaurus* (~1.11; Rieppel 1989), *Placodus* (~1.05; *Jiang et al., 2008*), and *Eusaurosphargis* (~0.89; *Scheyer et al., 2017*).

Tibia and fibula: The tibia is proximodistally slightly longer than the fibula and possesses proximally and distally expanded ends, with the proximal end slightly broader than the distal end (*Figure 5*). Posteroproximally, the tibia bears a proximodistally elongate, shallow facet for the fibula. The anterior and posterior margins of the tibia are gently concave. The fibula possesses an approximately straight posterior margin and a concave anterior margin (*Figure 5*). The proximal and distal ends of the fibula are slightly expanded, with the distal end being slightly broader than the proximal end. This is in contrast to *Largocephalosaurus*, in which the proximal end of the fibula is markedly broader than the distal end (*Li et al., 2014*).

Tarsals: Three tarsals are present in the left hindlimb of HFUT YZSB-19-109 – the largest one being the astragalus, the medium-sized representing the calcaneum, and the smallest element interpreted here as distal tarsal IV (*Figure 5*). The tarsals are sub-circular in outline, with the astragalus possessing minute notches anteriorly and posteriorly, and their exposed ventral surfaces are weakly concave. The morphology of the tarsals of HFUT YZSB-19-109 is similar to that in *Largocephalosaurus*, which, however, possesses four tarsals instead of three (astragalus, calcaneum and distal tarsals III and IV; *Li et al., 2014*).

Metatarsals: All five metatarsals are preserved in the left hindlimb of HFUT YZSB-19-109 (*Figure 5*). Metatarsal I is the proximodistally shortest metatarsal, being much broader proximally than distally. It is also the most robust of the metatarsals. Metatarsals II–V are slender, approximately hourglass-shaped in outline, with expanded proximal and distal ends. The proximodistal length of the metatarsals increases from metatarsal II–IV, with metatarsal IV being the longest of all metatarsals (similar to *Largocephalosaurus polycarpon*, but different from *L. qianensis*, in which metatarsal III is the longest; *Li et al., 2014*); the length of metatarsal V is comparable to that of metatarsal III. Metatarsal V is proportionally the most slender of the metatarsals (similar to the

condition in *L. qianensis*, but unlike in *L. polycarpon*, in which metatarsal IV is the most slender; *Li et al., 2014*).

Phalanges: The pedal phalanges are completely preserved in the left pes of HFUT YZSB-19-109 (*Figure 5*). The proximal phalanges of digits 1 and 2 are sub-rectangular in outline, whereas those of digits 3–5 are hourglass-shaped and have expanded proximal and distal ends. Distally, the phalanges become proximodistally shorter and sub-rectangular in outline. The phalangeal formula is 2-3-4-5-5 (unguals 2 and 3 are slightly displaced).

## Phylogenetic results

The phylogenetic analysis recovered 48 most parsimonious trees (MPTs) of 1008 steps each (CI = 0.272, RI = 0.620) (*Figure 8*). *Prosaurosphargis* was recovered as a member of Saurosphargidae, forming a polytomy with *Sinosaurosphargis* and *Largocephalosaurus*. Saurosphargidae is supported by the following three unambiguous synapomorphies: dorsal ribs transversely broadened and in antero-posterior contact with each other, forming closed 'rib-basket' (char. 135.1); lateral gastralia expanded and flat (char. 141.1); distal end of ulna distinctly expanded (char. 167.1).

Saurosphargidae was recovered within Sauropterygia, as the sister-group to the lineage leading to Eosauropterygia. The saurosphargid-eosauropterygian clade is supported by six unambiguous synapomorphies: frontal, butterfly-shaped with antero- and postero-lateral processes absent (ch. 29.0); long postorbital posterior process contacting squamosal (ch. 32.0); mandibular articulations displaced to a level distinctly behind occipital condyle (ch. 87.1); neural canal evenly proportioned (ch. 122.0); three sacral ribs (ch. 130.1); total number of carpal ossifications more than three (ch. 194.0).

*Pomolispondylus*, recently proposed as a sister-taxon of Saurosphargidae (*Cheng et al., 2022*), was recovered as the most basal member of the grade leading to Eosauropterygia (*Pomolispon-dylus* + eosauropterygian lineage supported by a single unambiguous synapomorphy: transverse processes of neural arches of the dorsal region relatively short [ch. 124.0]). The type (*Rieppel, 1998a*) and referred (*Wang et al., 2022*) specimens of *Hanosaurus* were not unambiguously recovered in a monophyletic group – both specimens were recovered alongside *Majiashanosaurus* in a polytomy at the base of Eosauropterygia (clade comprising *Hanosaurus*, *Majiashanosaurus*, and Eosauropterygia supported by a single unambiguous synapomorphy: osteoderms absent [ch. 143.0]). *Corosaurus* was recovered as the earliest-diverging member of Eosauropterygia, whereas *Wumengosaurus* was recovered outside of the clade comprising pachypleurosaurs, nothosaurs and pistosaurs.

The herbivorous sauropterygian *Atopodentatus* was recovered as the sister-taxon of placodonts (represented in our dataset by *Paraplacodus* and *Placodus*) within Placodontiformes, supported by four unambiguous synapomorphies: contact of the prefrontal and postfrontal excluding frontal from dorsal orbital margin (char. 21.1); interpterygoid vacuity absent (char. 81.1); splenial entering mandibular symphysis (char. 88.0); and femur internal trochanter well developed (char. 203.0).

*Palatodonta* was recovered as the sister-taxon of *Eusaurosphargis*. This clade is supported by one unambiguous synapomorphy – a small premaxilla (char. 196.1). The clade comprising *Palatodonta* + *Eusaurosphargis* was recovered as the sister-group to Sauropterygia within Sauropterygomorpha tax. nov. (see above). *Helveticosaurus* was recovered as the sister-group of Sauropterygomorpha; the clade comprising *Helveticosaurus* + Sauropterygomorpha is supported by the following six unambiguous synapomorphies: preorbital and postorbital regions of skull of subequal length (ch. 1.0); transverse processes of neural arches of the dorsal region distinctly elongated (ch. 124.1); scapula with a constriction separating a ventral glenoidal portion from a posteriorly directed dorsal wing (ch. 154.2); distal tarsal I absent (ch. 184.1); total number of tarsal ossifications less than four (ch. 192.1); total number of carpal ossifications two (char. 194.2).

Within Diapsida, the clade comprising *Helveticosaurus* + Sauropterygomorpha was recovered as forming a clade with Ichthyosauromorpha, Thalattosauria and Archosauromorpha, a result similar to that recovered in some other recent broad-scale analyses of diapsid phylogenetic interrelationships (*Chen et al., 2014a*; *Neenan et al., 2015*; *Martínez et al., 2021*; *Simões et al., 2022*). The three major marine reptile clades (Sauropterygomorpha, Ichthyosauromorpha and Thalattosauria), Archosauromorpha, and Testudines were recovered within a monophyletic Archelosauria supported by four unambiguous synapomorphies – frontal with distinct posterolateral processes (ch. 26.1), frontal anterior margins oblique, forming an angle of at least 30 degrees with long axis of the skull (ch. 27.1),

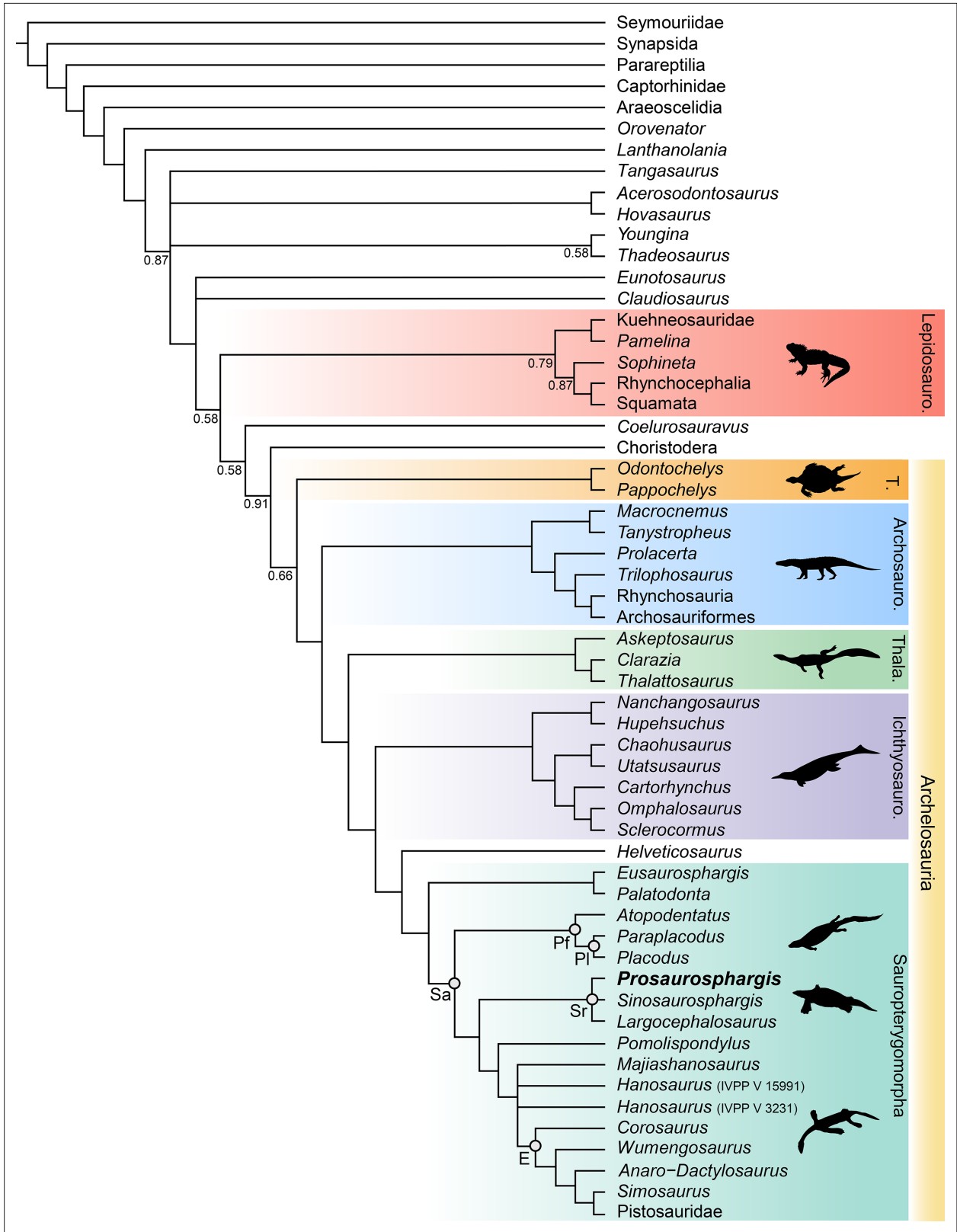

**Figure 8.** Phylogenetic relationships of *Prosaurosphargis yingzishanensis* within Diapsida. The 50% majority rule consensus of 48 most parsimonious trees (MPTs) obtained from analysis of the updated dataset of **Qiao et al., 2022**. Numbers below nodes indicate proportion of MPTs in which the node is recovered if it is lower than 1. Abbreviations: Archosauro., Archosauromorpha; E, Eosauropterygia; Ichthyosauro., Ichthyosauromorpha; Lepidosauro.,

*Figure 8 continued on next page*

Lepidosauromorpha; Pf, Placodontiformes; Pl, Placodontia; Sa, Sauropterygia; Sr, Saurosphargidae; T., Testudines (total-group); Thala., Thalattosauria. All silhouettes are from Phylopic (http://phylopic.org).

interclavicle anterior process or triangle conspicuously present (ch. 157.0), and upper temporal fossae present and distinctly smaller than the orbit (ch. 207.3).

## Discussion

### Marine reptile diversity of the Early Triassic Nanzhang-Yuan'an Fauna

*Prosaurosphargis* represents the stratigraphically oldest occurrence of Saurosphargidae, extending their fossil record back by approximately 3 Ma from the Middle (Pelsonian) to the Early (Olenekian) Triassic (*Figure 9*). Saurosphargids are thus the fourth major marine reptile lineage known from the Early Triassic Nanzhang-Yuan'an fauna, which also includes as many as seven species of hupehsuchians (*Chen et al., 2016a*; *Qiao et al., 2019*), one species of ichthyosauriforms (*Chen et al., 2013*), and three taxa representing the sauropterygian lineage leading to Eosauropterygia – *Hanosaurus*, *Lariosaurus sanxiaensis*, and *Pomolispondylus* (*Young, 1965*; *Rieppel, 1998a*; *Li and Liu, 2020*; *Cheng et al., 2022*; *Wang et al., 2022*). Measuring approximately 1.5 m in total body length, *Prosaurosphargis* is one of the larger marine reptiles known from this ecosystem, smaller only than a large unidentified eosauropterygian (body length of 3–4 m; *Chen et al., 2014b*, *Chen et al., 2016a*) and an indeterminate hupehsuchian (body length of ~2.3 m; *Qiao et al., 2019*). The presence in the Nazhang-Yuan'an fauna of several marine reptiles representing a broad range of body sizes (~0.25–4.00 m; *Chen et al., 2014b*; *Qiao et al., 2019*) and displaying various ecomorphological adaptations supports the view of a rapid diversification of predators in the immediate aftermath of the PTME and high predation pressure in shallow marine ecosystems in the Early Triassic (*Chen et al., 2014b*; *Chen et al., 2014c*; *Li and Liu, 2020*).

### Phylogeny of Sauropterygomorpha

A clade comprising *Palatodonta + Eusaurosphargis* and the lineage leading to *Helveticosaurus* are recovered as successive outgroups to Sauropterygia, with *Palatodonta + Eusaurosphargis* and Sauropterygia united within Sauropterygomorpha tax. nov. (see above) (*Figure 9*). The status of *Eusaurosphargis + Palatodonta* as the sister-group to Sauropterygia is corroborated by the presence in *Eusaurosphargis* of morphological features otherwise known exclusively in sauropterygians: a clavicle applied to the medial surface of the scapula and a pectoral fenestration (*Scheyer et al., 2017*). Furthermore, *Eusaurosphargis* and Sauropterygia also share a similar foot morphology with metatarsal I being proximodistally much shorter than metatarsal IV and metatarsal V being long and slender (see above). *Helveticosaurus* shares the presence of a skull with preorbital and postorbital regions subequal in length with *Eusaurosphargis* and *Palatodonta*, the presence of elongated dorsal transverse processes with *Eusaurosphargis*, placodonts and saurosphargids and the presence of a scapular constriction with *Eusaurosphargis* and sauropterygians. However, other anatomical features uniting it with Sauropterygomorpha include details of carpal and tarsal anatomy, which likely represent aquatic adaptations that might have evolved convergently in *Helveticosaurus* and aquatic representatives of Sauropterygomorpha (*Chen et al., 2014a*). Furthermore, *Helveticosaurus* lacks osteoderms, a feature present in early members of all major lineages within Sauropterygomorpha. Consequently, we interpret *Helveticosaurus* as a representative of a lineage closely related to Sauropterygomorpha but lying outside of it, that likely convergently evolved an aquatic lifestyle.

Our phylogenetic analysis does not recover *Palatodonta* from the Middle Triassic of the Netherlands as a sister-taxon of placodonts within Placodontiformes as was previously proposed (*Neenan et al., 2013*). Instead, *Palatodonta* is recovered as the sister-taxon of *Eusaurosphargis*. The *Palatodonta + Eusaurosphargis* clade recovered in our phylogenetic analysis is supported by one unambiguous

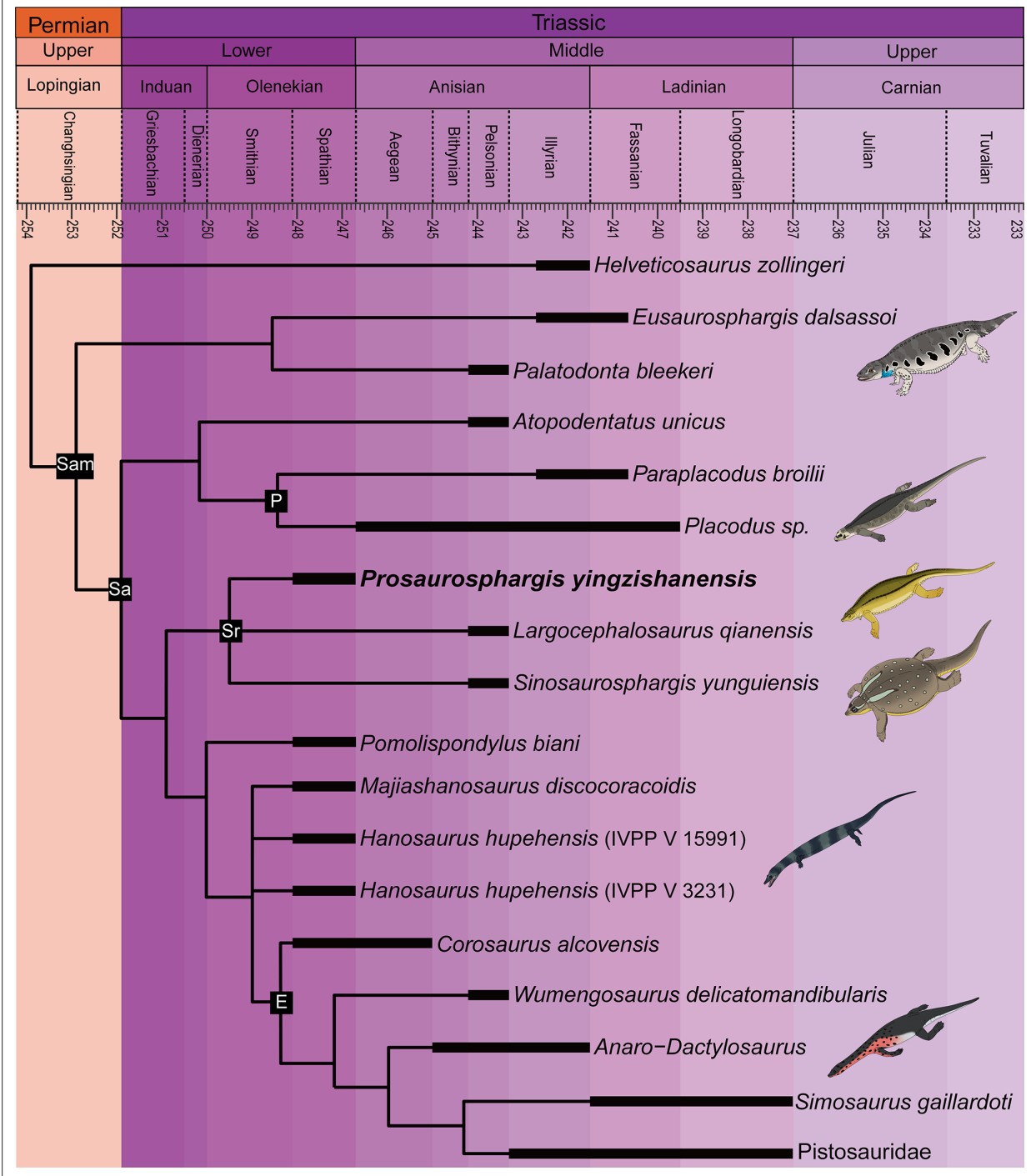

**Figure 9.** Time-scaled phylogenetic tree of Sauropterygomorpha. Abbreviations: E, Eosauropterygia; Pf, Placodontiformes; Pl, Placodontia; Sa, Sauropterygia; Sam, Sauropterygomorpha; Sr, Saurosphargidae.

synapomorphy – a small (anteroposteriorly short) premaxilla (ch. 196.0). In addition, both taxa also share anteriorly positioned external nares (ch. 197.0). Both of these character states are considered as typical for terrestrial taxa and contrast with the enlarged (anteroposteriorly elongate) premaxillae and posteriorly displaced external narial openings characteristic for marine reptiles (*Chen et al., 2014a*). A terrestrial lifetyle was previously proposed for *Eusaurosphargis* on the basis of manus and pes anatomy and bone microanatomy (*Scheyer et al., 2017*) and the femur of *Eusaurosphargis* is longer than its humerus (humerus:femur ratio ~0.89; *Scheyer et al., 2017*), which indicates hindlimb dominance

characteristic of a terrestrial lifestyle (*Motani and Vermeij, 2021*). All this evidence strongly suggests that *Palatodonta* and *Eusaurosphargis* were terrestrial reptiles, likely representing the morphology of the last common terrestrial ancestor of Sauropterygia. *Palatodonta* is known from a single isolated skull (*Neenan et al., 2013*), whereas *Eusaurosphargis* is represented by two specimens with well preserved and largely complete postcrania, but only partially preserved skulls (*Nosotti and Rieppel, 2003*; *Scheyer et al., 2017*). Because postcranial remains referable to *Eusaurosphargis* were also reported from the type locality of *Palatodonta*, including a *Palatodonta*-like dentary preserved in close association with typical *Eusaurosphargis*-like vertebrae (*Scheyer et al., 2019*; *Willemse et al., 2019*), it is very likely that *Palatodonta* is a junior synonym of *Eusaurosphargis*, but the discovery of well-preserved skulls with associated postcranial elements of both of these taxa are needed to further test this hypothesis.

*Atopodentatus* and placodonts are recovered within Placodontiformes and this grouping is supported by four unambiguous synapomorphies (see above). *Atopodentatus* and the early-diverging placodonts *Placodus* and *Paraplacodus* possess a humerus which is longer than the femur, indicating a high level of adaptation to an aquatic lifestyle (*Motani and Vermeij, 2021*), but they also possess a massive femoral fourth trochanter and an ilium with a well-developed iliac blade (*Jiang et al., 2008*; *Cheng et al., 2014*). These features indicate that the hindlimbs in both *Atopodentatus* and placodonts were likely still important in locomotion at the bottom of the sea floor and/or on shore in a marginal marine environment and suggest a slightly lower degree of adaptation to an aquatic lifestyle in placodontiforms than in saurosphargids and eosauropterygians, in which the fourth trochanter is more reduced (*Rieppel, 2000a*; *Li et al., 2014*). The sister-group relationships of *Atopodentatus* and placodonts might also explain the absence of placodont fossils in Early Triassic fossil horizons worldwide. It is possible that the lineage leading to placodonts was represented in the Early Triassic by reptiles morphologically more similar to *Atopodentatus* than to placodonts and that the specialised placodont body plan did not evolve until the Middle Triassic. New discoveries of Early Triassic sauropterygians are likely to introduce new morphological data needed to test this hypothesis.

*Pomolispondylus* is not recovered as a sister-taxon of Saurosphargidae within Saurosphargiformes (contra *Cheng et al., 2022*), but as the most basal member of a grade of sauropterygians leading to Eosauropterygia. Such a phylogenetic position is supported by the presence in *Pomolispondylus* of dorsal transverse processes that are relatively short mediolaterally and broad anteroposteriorly, more similar in proportions to the dorsal transverse processes of *Lariosaurus sanxiaensis* (*Li and Liu, 2020*) and *Hanosaurus* (*Wang et al., 2022*) – two other representatives of the grade leading to Eosauropterygia – than to the mediolaterally broad and anteroposteriorly narrow dorsal neural spines of saurosphargids (*Figure 3*; *Li et al., 2011*; *Li et al., 2014*). Furthermore, *Pomolispondylus* possesses rows of rudimentary osteoderms on its body flanks, which are much more reduced than those present in *Eusaurosphargis*, early-diverging placodonts and saurosphargids (*Klein and Scheyer, 2013*; *Li et al., 2014*; *Scheyer et al., 2017*). Therefore, the osteoderms in *Pomolispondylus* likely represent a late stage of osteoderm reduction in the lineage leading to Eosauropterygia, rather than the first stages of osteoderm development in the saurosphargid lineage.

The type (*Young, 1972*; *Rieppel, 1998a*) and referred (*Wang et al., 2022*) specimens of *Hanosaurus* and *Majiashanosaurus* (*Jiang et al., 2014*) are also recovered in the paraphyletic grade leading to Eosauropterygia, but in a position more derived than *Pomolispondylus*. This result is in contrast to previous studies which recovered these taxa as either the outgroup to Saurosphargidae + Sauropterygia (*Hanosaurus*; *Wang et al., 2022*), pachypleurosaurs (*Rieppel, 1998a*; *Neenan et al., 2015*; *Lin et al., 2021*) or successive outgroups to a clade comprising Pachypleurosauria + Nothosauroidea to the exclusion of Pistosauroidea (*Li and Liu, 2020*). Suprisingly, the type specimen of *Hanosaurus* is not unambiguously recovered in a clade with the referred specimen. Taxonomic distinction of both specimens is supported by the fact that the anteriorly and posteriorly weakly emarginated coracoid of the type specimen of *Hanosaurus* (*Figure 7G and H*; *Rieppel, 1998a*; pers. obs. of IVPP V 3231) more closely resembles that of *Corosaurus* than the sub-oval coracoid present in the referred specimen of *Hanosaurus* (*Figure 7I*; *Wang et al., 2022*), which likely does not represent *Hanosaurus*, but is probably closely related or even referable to *Lariosaurus sanxiaensis* (*Figure 7J*; *Chen et al., 2016b*; *Li and Liu, 2020*). *Corosaurus* is recovered as the earliest-diverging eosauropterygian, a result similar to that obtained by *Rieppel, 1994* and *Li and Liu, 2020*, but in contrast to some other phylogenetic analyses, which recovered it as a pistosauroid (*Rieppel, 1998b*; *Wang et al., 2022*) or the earliest-diverging

eusauropterygian (*Lin et al., 2021*). *Wumengosaurus* is recovered as the sister-taxon of the clade comprising pachypleurosaurs, nothosaurs and pistosaurs, in contrast to a recent phylogenetic analysis which recovered it wtihin pachypleurosaurs (*Xu et al., 2022*), but similar to the phylogenetic results obtained by *Wu et al., 2011* that recovered *Wumengosaurus* as the outgroup to a clade comprising pachypleurosaurs and nothosaurs.

The results of our phylogenetic analysis differ significantly from the results of a phylogenetic analysis recently published by *Wang et al., 2022*, in which *Hanosaurus* was recovered as the sister-group to a clade comprising Saurosphargidae + Sauropterygia within a monophyletic Sauropterygiformes. However, we believe that our phylogenetic analysis presents a more accurate topology of sauropterygians and their relatives for the following reasons. *Wang et al., 2022* used 16 outgroup (non-sauropterygiform) taxa (15 marine reptiles and a single terrestrial reptile), representing five major reptile lineages, whereas our study included 40 outgroup (non-sauropterygomorph) taxa (19 terrestrial and 21 aquatic reptiles), representing 23 major reptile lineages. The inclusion of only a single terrestrial outgroup taxon – *Youngina* – in the analysis of *Wang et al., 2022* is problematic because *Youngina* likely represents a taxon rather distantly related to the Mesozoic marine reptile clade which includes Sauropterygia (*Figure 8*; see also *Simões et al., 2022*). Distantly related taxa likely share fewer character states with derived taxa (homoplasy accumulation through time), so a phylogenetic analysis containing a single, distantly related outgroup taxon will likely fail to adequately capture important character transformation sequences, in contrast to a phylogenetic analysis in which outgroups are more comprehensively sampled (*Wilberg, 2015*). Furthermore, the phylogenetic analysis of *Wang et al., 2022* used 181 morphological characters, in contrast to 221 characters included in our study. Greater character sampling has been demonstrated as an important factor increasing the accuracy of phylogenetic reconstructions (*Wiens, 2006*), which also favours the results obtained by our analysis.

The different topologies recovered by both studies are likely also partially a consequence of differences in character scoring. For example, *Wang et al., 2022* scored the humerus of *Hanosaurus* as 'rather straight' (plesiomorphic state), similar to the humerus of *Youngina* and other non-sauropterygiform marine reptiles. However, in our opinion, the humerus morphology of *Hanosaurus*

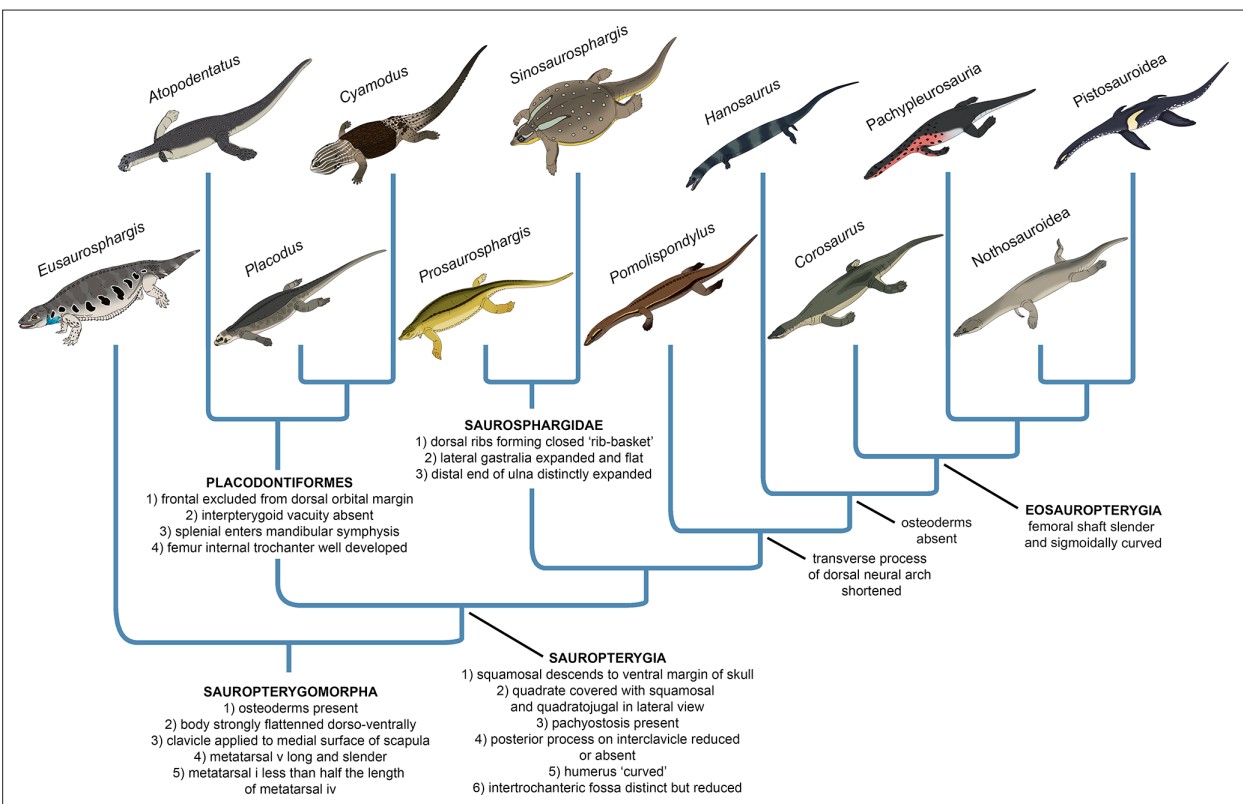

**Figure 10.** Evolution of the sauropterygian body plan. Simplified phylogeny of Sauropterygomorpha with key anatomical traits (synapomorphies reconstructed from the phylogenetic analysis) indicated for important nodes.

matches the typical 'curved' morphology (derived state) characteristic of saurosphargids, placodonts and the majority of eosauropterygians (*Rieppel, 2000a*). *Wang et al., 2022* also scored *Hanosaurus* into an updated version of the phylogenetic matrix of *Neenan et al., 2013*. *Hanosaurus* was recovered as the earliest-diverging sauropterygiform in this analysis as well, but its humerus was also scored as plesiomorphic ('rather straight') in the character-taxon matrix. Interestingly, this updated analysis of *Neenan et al., 2013* included a more comprehensive outgroup (non-sauropterygiform) sample (12 terrestrial and 5 aquatic taxa, representing 12 major reptile lineages) than the analysis of *Wang et al., 2022* and recovered *Eusaurosphargis* and *Helveticosaurus* as successive sister-groups to Sauropterygia – a result similar to the one obtained in this study (*Figure 9*).

## The early evolutionary assembly of the sauropterygian body plan

Our phylogenetic analysis suggests that *Eusaurosphargis* and *Palatodonta* likely represent the morphology of the last common terrestrial ancestor of sauropterygians, indicating it possessed well-developed dermal armour and the characteristic pectoral girdle and pes morphology that underwent further modifications in Sauropterygia. The topology recovered by our phylogenetic analysis demonstrates that the early evolution of sauropterygians first involved diversification within a shallow marine environment and exploration of various food resources, as evidenced by the disparate ecologies exhibited by *Atopodentatus* (herbivore), placodonts (durophages), saurosphargids, and early-diverging members of the eosauropterygian lineage (likely feeding on fish and invertebrates). Three key episodes can be identified in the evolution of the eosauropterygian body plan (*Figure 10*). The first, represented by *Pomolispondylus*, involved a reduction of osteoderms and shortening of the transverse processes of the dorsal neural spines, features well-developed in *Eusaurosphargis*, placodonts and saurosphargids. *Majiashanosaurus* and the referred specimen of *Hanosaurus* exemplify the second episode, in which osteoderms underwent complete reduction. The type specimen of *Hanosaurus* represents the earliest stage of the evolution of the characteristic eosauropterygian pectoral girdle morphology with an anteriorly and posteriorly emarginated coracoid that ultimately allowed eosauropterygians to become efficient, paraxial swimmers. The presence of the stratigraphically oldest saurosphargid (*Prosaurosphargis*), stratigraphically oldest representatives of the lineage leading to Eosauropterygia (*Pomolispondylus*, *Hanosaurus*, *Majiashanosaurus*), and the earliest-diverging placodontiform *Atopodentatus* in the Early–Middle Triassic of South China (*Figure 9*) indicates that sauropterygians likely originated and underwent rapid diversification in South China in the aftermath of the end-Smithian extinction, similar to ichthyosauromorphs (*Motani et al., 2017*; *Moon and Stubbs, 2020*), but well-constrained stratigraphic data for early sauropterygians are needed to further test this hypothesis.

Our phylogenetic analysis indicates the important role of body armour in sauropterygian evolution (*Figure 10*). Dermal armour was likely an important preadaptation that allowed colonisation of the shallow marine realm by a *Eusaurosphargis*-like ancestor, enabling it to counteract buoyancy and walk on the bottom of the shallow sea in search of food (*Houssaye, 2009*). Elaboration of the dermal armour occurred in the shallow marine placodonts and saurosphargids, perhaps as a response to predation pressure (*Liu et al., 2014*; *Chen et al., 2014b*; *Qiao et al., 2019*). The reduction and complete loss of dermal armour then occurred in the lineage leading to Eosauropterygia, and was likely associated with the evolution of active predation, an efficient swimming style and increasing adaptation to a pelagic lifestyle. The evolution of the sauropterygian body plan demonstrates striking parallels with the evolution of the body plan of another important group of Mesozoic marine reptiles – the ichthyosaurs. Early-diverging representatives of Ichthyosauromorpha – hupehsuchians and omphalosaurids – also possessed a covering of osteoderms superficially similar to that present in *Eusaurosphargis*, early-diverging placodonts, and saurosphargids (*Chen et al., 2014b*; *Jiang et al., 2016*; *Qiao et al., 2022*). In the ichthyosauromorph lineage, the osteoderm covering was completely lost in *Chaohusaurus*, a basal ichthyosauriform that evolved an anguilliform mode of swimming and most likely a pelagic lifestyle (*Motani et al., 1996*; *Nakajima et al., 2014*; *Qiao et al., 2022*). This indicates that the evolutionary reduction of dermal armour in both sauropterygians and ichthyosauromorphs followed a similar pattern, probably in response to increasing adaptation to an aquatic lifestyle. These evolutionary parallels seem to demonstrate that the dermal body armour could have been a possible prerequisite (preadaptation) for the invasion of the shallow marine realm in different diapsid clades, which allowed for buoyancy reduction and exploration of the shallow marine environment in search

of food. Fossils of terrestrial relatives of ichthyosauromorphs and thalattosaurs are needed to further test this evolutionary scenario.

## Phylogenetic interrelationships within Diapsida

Our phylogenetic analysis is thus far one of only two phylogenetic analyses based on a morphology-only dataset of Diapsida that recovers a close relationship between Archosauromorpha and Testudines within a monophyletic Archelosauria (*Figure 8*) (see also *Simões et al., 2022*). A close phylogenetic relationship between Archosauromorpha and Testudines has been strongly supported by molecular data for the last twenty years, but was until recently not recovered by any phylogenetic analysis based entirely on morphological data, in which turtles were usually recovered as more closely related to lepidosauromorphs than archosauromorphs (reviewed in *Lyson and Bever, 2020*). Furthermore, our analysis recovers *Eunotosaurus* from the Permian of South Africa outside of Sauria (*Figure 8*), which is in agreement with the results obtained by *Simões et al., 2022*, but in contrast to all other recent phylogenetic analyses focussing on the phylogenetic interrelationships among Reptilia, which recovered *Eunotosaurus* as a stem turtle (*Lyson et al., 2013*; *Bever et al., 2015*; *Li et al., 2018*; *Schoch and Sues, 2018*). This indicates that the characteristic morphological features of the skull and postcranial skeleton shared between *Eunotosaurus* and early turtles, such as elongate vertebrae and broadened ribs, evolved convergently in both taxa. Archelosauria are supported in our analysis by

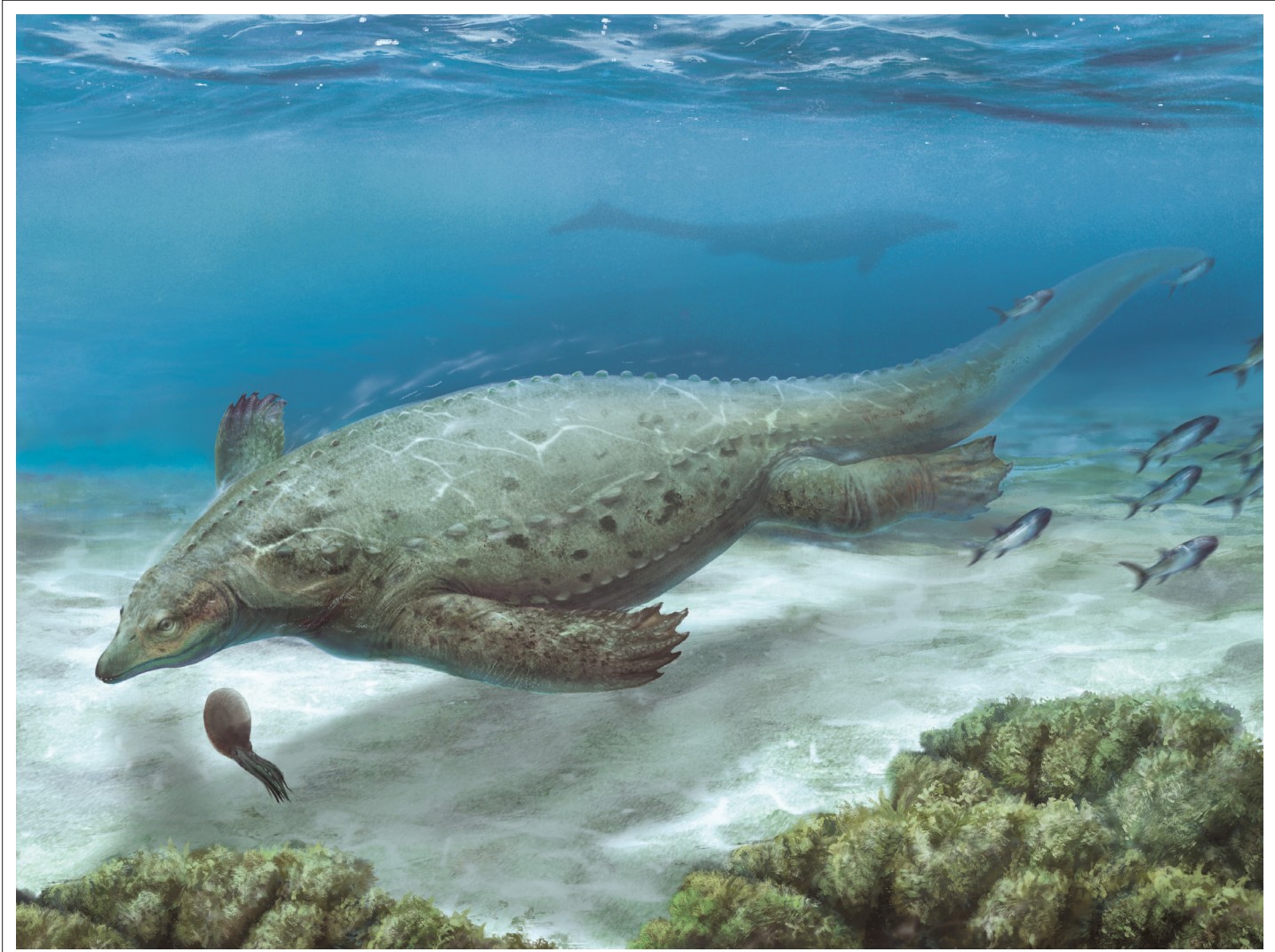

**Figure 11.** Life reconstruction of *Prosaurosphargis yingzishanensis* depicted in the Early Triassic shallow marine environment of the Nanzhang-Yuan'an region, Hubei Province, South China.

four unambiguous synapomorphies associated with the morphology of the skull and shoulder girdle (see above). This demonstrates that innovations of the cranium linked with the evolution of sensory organs and the feeding apparatus, as well as changes in locomotion, perhaps underlie the evolutionary success of archelosaurian reptiles.

## Conclusions

The new saurosphargid *Prosaurosphargis yingzishanensis* gen. et sp. nov. from the Early Triassic of South China (*Figure 11*) represents the earliest reported occurrence of Saurosphargidae, extending their temporal range back by 3 Ma. An updated phylogenetic analysis of Diapsida recovers saurosphargids as nested within Sauropterygia, forming a clade with Eosauropterygia to the exclusion of Placodontia. A clade comprising *Eusaurosphargis* and *Palatodonta* forms the sister-group to Sauropterygia within Sauropterygomorpha tax. nov. and their morphology likely represents the morphology of the last common terrestrial ancestor of Sauropterygia. The herbivorous sauropterygian *Atopodentaus* is recovered within Placodontiformes, whereas *Pomolispondylus*, *Hanosaurus* and *Majishanosaurus* form a grade at the base of Eosauropterygia, with the type and referred specimens of *Hanosaurus* likely representing distinct taxa. Our new phylogenetic hypothesis indicates sauropterygians originated and diversified in South China in the aftermath of the Permo-Triassic mass extinction event and suggests an important role of dermal armour in their early evolutionary history. Three major marine reptile clades – Sauropterygomorpha, Ichthyosauromorpha and Thalattosauria – are recovered within Archelosauria, together with Archosauromorpha and Testudines. Our study demonstrates the importance of including not only a broad sample of outgroup taxa in phylogenetic analyses, but also choosing their stratigraphically oldest and/or anatomically most plesiomorphic representatives as operational taxonomic units, in order to accurately reconstruct the phylogenetic relationships between major extant and extinct reptilian lineages.

## Materials and methods

In order to investigate the phylogenetic position of *Prosaurosphargis yingzishanensis*, specimen HFUT YZSB-19-109 was scored into a modified version of a data matrix focusing on the phylogenetic interrelationships between the major groups of diapsid reptiles published by *Qiao et al., 2022*, which in itself is a modified version of the data matrix previously published by *Jiang et al., 2016* and *Chen et al., 2014a*. The data matrix contains 57 OTUs (Operational Taxonomic Units) scored for a total of 221 characters – characters 1–220 are the original characters of *Qiao et al., 2022* and character 221 was adapted from *Li et al., 2014* (ch. 88) (*Source data 1*). In addition to *Prosaurosphargis*, 10 OTUs were added to the original dataset of *Qiao et al., 2022*: *Eunotosaurus africanus* (scored after *Cox, 1969*; *Gow, 1997*; *Lyson et al., 2013*; *Lyson et al., 2016*; *Bever et al., 2015*), *Pappochelys rosinae* (*Schoch and Sues, 2015*; *Schoch and Sues, 2018*); *Hanosaurus hupehensis* (type specimen) (*Rieppel, 1998a* and personal observation of specimen IVPP V 3231), *Hanosaurus hupehensis* (referred specimen) (*Wang et al., 2022*), *Majiashanosaurus discocoracoidis* (*Jiang et al., 2014*), *Corosaurus alcovensis* (*Storrs, 1991*; *Rieppel, 1998b*), *Atopodentatus unicus* (*Cheng et al., 2014*; *Li et al., 2016*), *Palatodonta bleekeri* (*Neenan et al., 2013*), *Paraplacodus broilii* (*Peyer, 1935*; *Rieppel, 2000b*), and *Pomolispondylus biani* (*Cheng et al., 2022*). This was done in order to include the majority of currently known Early Triassic sauropterygians in the data matrix and increase the sampling of early-diverging representatives of the main sauropterygian lineages, as well as their potential sister-groups. Furthermore, the holotype and referred specimen of *Sclerocormus*, included as separate OTUs in the dataset of *Qiao et al., 2022*, were merged into a single OTU in the current analysis.

Parsimony analysis of the data matrix was performed in TNT 1.5 (*Goloboff and Catalano, 2016*) using a Traditional Search algorithm (random seed = 1, replications of Wagner trees = 1000, number of trees saved per replication = 10), followed by an additional round of TBR branch-swapping. All characters were treated as equally weighted and unordered.

## Acknowledgements

We thank PM Sander for early discussions, W Lin for sharing photographs of the *Hanosaurus* holotype, BY Xiang, XJ Liu and the HFUT Paleontology Lab members for assistance in the field, and the Willi Hennig Society for making the programme TNT publicly available. F Huang, L Li and T Sato skilfully

prepared the fossil specimen. T Hollmann produced the skeletal reconstruction in *Figure 2B* and the sauropterygian cartoons in *Figures 9 and 10*, and Z Han provided the life reconstruction in *Figure 11*. The senior editor GH Perry, reviewing editor N Ibrahim, reviewer MJ Benton and another annonymous reviewer provided constructive comments that helped to improve the manuscipt.

## Additional information

### Funding

| Funder | Grant reference number | Author |
| --- | --- | --- |
| National Natural Science Foundation of China | 42172026 | Jun Liu |
| National Natural Science Foundation of China | 41772003 | Jun Liu |
| National Natural Science Foundation of China | 42202006 | Andrzej S Wolniewicz |
| National Natural Science Foundation of China | 41902104 | Yuefeng Shen |
| National Natural Science Foundation of China | 41807333 | Yuanyuan Sun |
| Chengdu Center of China Geological Survey | Liu Baojun Funding | Yuanyuan Sun |
| China Postdoc Council | International Postdoctoral Exchange Fellowship Program | Andrzej S Wolniewicz |

The funders had no role in study design, data collection and interpretation, or the decision to submit the work for publication.

### Author contributions

Andrzej S Wolniewicz, Software, Formal analysis, Funding acquisition, Validation, Investigation, Visualization, Methodology, Writing - original draft, Writing – review and editing; Yuefeng Shen, Formal analysis, Funding acquisition, Validation, Investigation, Methodology, Writing - original draft; Qiang Li, Formal analysis, Validation, Investigation, Visualization, Methodology; Yuanyuan Sun, Investigation; Yu Qiao, Software, Formal analysis, Investigation, Methodology; Yajie Chen, Software, Investigation, Visualization; Yi-Wei Hu, Investigation, Visualization; Jun Liu, Conceptualization, Resources, Data curation, Formal analysis, Supervision, Funding acquisition, Validation, Investigation, Methodology, Project administration, Writing – review and editing

### Author ORCIDs

Andrzej S Wolniewicz ⓘ https://orcid.org/0000-0002-6336-8916
Jun Liu ⓘ https://orcid.org/0000-0001-7859-5209

### Decision letter and Author response

Decision letter https://doi.org/10.7554/eLife.83163.sa1
Author response https://doi.org/10.7554/eLife.83163.sa2

## Additional files

### Supplementary files

MDAR checklist

Source data 1. Character-taxon matrix used in the phylogenetic analysis.

## Data availability

Specimen HFUT YZSB-19-109 is housed in the collections of the Geological Museum, Hefei University of Technology, Hefei, China and available for examination upon request to JL. The phylogenetic data matrix used in this study is available in *Source data 1*.

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
