## [Editor Report]

This significant new study is built around a remarkable fossil of a new genus and species of armoured marine reptile from the Early Triassic of China. More importantly, this paper also adds to our understanding of the diversification of reptiles during the Triassic and also sheds light on the interrelationships of a wide range of important groups. The authors present a solid phylogenetic analysis of sauropterygians, which reveals a possible new clade, Sauropterygomorpha, which is recovered close to Archosauromorpha. The authors suggest that the latter, as well as the Testudines and Ichthyosauromorpha, belong to a clade that had previously only been recovered using phylogenomic data, the Archelosauria. This paper will be of interest to a broad range of scientists, including palaeontologists, herpetolgists, and evolutionary biologists.

---

## [Decision Letter]

**Decision letter after peer review:**

Thank you for submitting your article "An armored marine reptile from the Early Triassic of South China" for consideration by *eLife*. Your article has been reviewed by 2 peer reviewers, and the evaluation has been overseen by a Reviewing Editor and George Perry as the Senior Editor. The following individual involved in review of your submission has agreed to reveal their identity: Michael Benton (Reviewer #1).

Essential revisions:

As you can see, there was considerable enthusiasm for the work, and there are only a few comments:

1) Please implement the changes suggested by reviewer 1 (mostly typos and improvements of figure layout etc).

2) Add a few more comments on the recently published paper "Wei Wang, Qinghua Shang, Long Cheng, Xiao-Chun Wu, Chun Li, Ancestral body plan and adaptive radiation of sauropterygian marine reptiles, iScience, 25(12) 2022, https://doi.org/10.1016/j.isci.2022.105635." (see comments by reviewer 2).

*Reviewer #1 (Recommendations for the authors):*

There are two presentations on the geology of the locality and fauna lines 121-132 and

170-197 (stratigraphic note). I would combine the Stratigraphic note with the short existing section on Geological setting to avoid repetition and show you deal with the age of the fauna before describing the fossil.

Figure 1: Make sure the lettering is large enough to be read – especially on the stratigraphic chart.

105: different.

278, 280, 285: you refer to plates in Huene (1936) as taf. [=Tafel], but pl. at line 422; I think I'd use 'pl.' in each case, translating into English. Also, I see pl. XIII instead of pl. 13, at line 422; do we use Latin numbers sometimes and Arabic numbers at other times? I'd standardise to Arabic.

329: 1836 = 1936

498: 1836 = 1936

579: linage = lineage

601: spelnial = splenial

762: whch = which

801: monophyletic

Incomplete references: Hirasawa et al. 2013; Jiang et al. 2016; Li and Liu 2020; Moon and Stubbs 2020; Scheyer et al. 2017; Xu et al. 2022 – these all lack article numbers. You need to train Endnote or Procite properly!

*Reviewer #2 (Recommendations for the authors):*

I know that the publication of Wei Wang, Qinghua Shang, Long Cheng, Xiao-Chun Wu, Chun Li, Ancestral body plan and adaptive radiation of sauropterygian marine reptiles, iScience, 25(12) 2022, https://doi.org/10.1016/j.isci.2022.105635 was only very recently published and it is already cited in the manuscript. However, Wolniewicz et al. have to discuss the phylogenetic relationships found in this paper as well as the implications drawn on ancestral body shape in Wei Wang et al. 2022 in more detail in their manuscript since these authors analyse the same marine reptile groups but came up with different hypotheses. I know that one major problem is the different interpretation of Hanosaurus and its referred specimen and Lariosaurus sanxian. Wolniewicz et al. already included a discussion of the referred specimen of Hanosaurus and justified by morphological arguments their different interpretation. This is all fine but nevertheless the phylogenetic conclusions needs to be compared and discussed/need some elaborations. A good way to do so will be to compare/discuss the charatcers which led to the one and the other result/phylo. interrelationships.

And of course one have always to consider that many taxa included in all these analyses are based (at least partially) only on incomplete single specimens. This is very problematic because we know from other taxa represented by several individuals that marine reptiles are prone to ontogenetic-intraspecific and sexual dimorphism. Thus drawing conclusions and erecting new taxa on the basis of single and incomplete individuals is problematic. However, I understand that this is the way it goes and only new material can verify or falsify conclusions drawn on those specimens.

1. I would suggest to change or make an addendum to the title since your paper indicates more than just another new reptile -and its implications for new phylo hypos or so (lines 1–2).

2. difrerent (line 105).

3. I do not understand this character since most of the early marine reptiles (e.g., Lario sanxian., Hano, Majiash. etc) do NOT have osteoderms (line 150).

4. From your figure it is also possible that this specimen has 4 to 5 sacral vert/ribs, since the rib anterior to your sr1 also is distally expanded as is the rib posterior to your sr3. please explain why these are not sr (line 299).

5. [a rugose dorsolateral surrface] please show/add in figure (line 310).

6. aff Eusaurosphargis from Winterswijk does have uncinate processes on dorsal ribs but maybe this depends on anatomical position and ontogeny ? (line 333)

7. [‘rib-basket’] can you please visualize this by a sketch not totally clear what you mean (line 338).

8. As far as I understand the coracoid is an important element to distinguish taxa please add an overview figure (photo or sketch) of the coracoids of all impoartnat taxa (line 413).

9. Why ‘Lariosaurus’/in quotation marks (line 419).

10. This [a weak connection between dorsal neural arches and centra] is one characteristic of Sauropterygia , summarized in Rieppel 2000; in addition vertebral ossification follows in different groups different patterns, it is thus not necessarily a good indicator for ontogeny (line 549).

11. You must discuss and compare your [phylogenetic] results with those of Wang et al. 2022 (line 560).

12. ploytomy (line 563).

13. Did you also include in another version Cyamodontidae/heavily armored placodonts ? I am wondering if this would make any change (line 570).

14. "small" etc is a difficult character which should be defined (for example give a ratio or smaller than XY) (line 604).

15. Sauopterygomorpha (line 667).

16. Please add reference for this statement (line 676).

17. posteriory (line 677).

18. ? to my knowledge there is no bone histological study on Eusaurosphargis so far; microanatomy based on Ct data are mentioned in Scheyer et al. 2017 (line 679).

19. Unclear statement please rephrase to be clearer (line 704).

20. Please add results and discuss those from Wang et al. 2022 which have a completely different hypo (line 748).

21. demonstreates (line 753).

22. eosauropterygia (line 767).

23. unclear/why (lines 777–778; 792–793).

24. Please shortly discuss some of the characters which lead to this result (line 799).

---

## [Author Response]

Essential revisions:As you can see, there was considerable enthusiasm for the work, and there are only a few comments:1) Please implement the changes suggested by reviewer 1 (mostly typos and improvements of figure layout etc).

Please see responses to Reviewer 1 comments below.

2) Add a few more comments on the recently published paper "Wei Wang, Qinghua Shang, Long Cheng, Xiao-Chun Wu, Chun Li, Ancestral body plan and adaptive radiation of sauropterygian marine reptiles, iScience, 25(12) 2022, https://doi.org/10.1016/j.isci.2022.105635." (see comments by reviewer 2).

Please see responses to Reviewer 2 comments below.

Reviewer #1 (Recommendations for the authors):There are two presentations on the geology of the locality and fauna lines 121-132 and170-197 (stratigraphic note). I would combine the Stratigraphic note with the short existing section on Geological setting to avoid repetition and show you deal with the age of the fauna before describing the fossil.

The stratigraphic note (lines 170–197) was integrated with the text of the Geological Background section (lines 121–132), in line with the reviewers suggestions.

Figure 1: Make sure the lettering is large enough to be read – especially on the stratigraphic chart.

The font size in Figure 1 was increased according to the reviewers suggestion and a revised Figure 1 was included in the revised version of the manuscript.

105: different.278, 280, 285: you refer to plates in Huene (1936) as taf. [=Tafel], but pl. at line 422; I think I'd use 'pl.' in each case, translating into English. Also, I see pl. XIII instead of pl. 13, at line 422; do we use Latin numbers sometimes and Arabic numbers at other times? I'd standardise to Arabic.329: 1836 = 1936498: 1836 = 1936579: linage = lineage601: spelnial = splenial762: whch = which801: monophyletic

The references to specific figures and plates were removed from the in-text citations to improve consistency and clarity of the manuscript. We thank the reviewer for correcting the typos. All of them were corrected in the revised manuscript.

Incomplete references: Hirasawa et al. 2013; Jiang et al. 2016; Li and Liu 2020; Moon and Stubbs 2020; Scheyer et al. 2017; Xu et al. 2022 – these all lack article numbers. You need to train Endnote or Procite properly!

We thank the reviewer for pointing out these incomplete references. The article numbers were added to these references in the revised manuscript.

Reviewer #2 (Recommendations for the authors):I know that the publication of Wei Wang, Qinghua Shang, Long Cheng, Xiao-Chun Wu, Chun Li, Ancestral body plan and adaptive radiation of sauropterygian marine reptiles, iScience, 25(12) 2022, https://doi.org/10.1016/j.isci.2022.105635 was only very recently published and it is already cited in the manuscript. However, Wolniewicz et al. have to discuss the phylogenetic relationships found in this paper as well as the implications drawn on ancestral body shape in Wei Wang et al. 2022 in more detail in their manuscript since these authors analyse the same marine reptile groups but came up with different hypotheses. I know that one major problem is the different interpretation of Hanosaurus and its referred specimen and Lariosaurus sanxian. Wolniewicz et al. already included a discussion of the referred specimen of Hanosaurus and justified by morphological arguments their different interpretation. This is all fine but nevertheless the phylogenetic conclusions needs to be compared and discussed/need some elaborations. A good way to do so will be to compare/discuss the charatcers which led to the one and the other result/phylo. interrelationships.And of course one have always to consider that many taxa included in all these analyses are based (at least partially) only on incomplete single specimens. This is very problematic because we know from other taxa represented by several individuals that marine reptiles are prone to ontogenetic-intraspecific and sexual dimorphism. Thus drawing conclusions and erecting new taxa on the basis of single and incomplete individuals is problematic. However, I understand that this is the way it goes and only new material can verify or falsify conclusions drawn on those specimens.

Please see detailed responses below.

1. I would suggest to change or make an addendum to the title since your paper indicates more than just another new reptile -and its implications for new phylo hypos or so (lines 1–2).

We thank the reviewer for this suggestion. The title of the manuscript was modified into “An armoured marine reptile from the Early Triassic of South China and its phylogenetic and evolutionary implications”.

2. difrerent (line 105).

Corrected to ‘different’ (typo also noticed by Reviewer 1).

3. I do not understand this character since most of the early marine reptiles (e.g., Lario sanxian., Hano, Majiash. etc) do NOT have osteoderms (line 150).

Establishing the presence of osteoderms as a synapomorphy of Sauropterygomorpha (defined as the last common ancestor of *Eusaurosphargis* + *Pistosaurus* and all of its descendants) results from our phylogenetic analysis in TNT 1.5, which reconstructed the presence of osteoderms as a synapomorphy of the Sauropterygomorpha node. This is why this character state is included in the diagnosis of the clade. Please note that *Eusaurosphargis*, *Placodus*, Saurosphargidae and *Pomolispondylus* – the most basal member of the lineage leading to Eosauropterygia according to our phylogenetic hypothesis – all share the presence of osteoderms, supporting it as a synapomorphy of Sauropterygomorpha (osteoderms were likely convergently lost in *Atopodentatus*, *Paraplacodus* and the lineage leading to Eosauropterygia crownward of *Pomolispondylus*).

4. From your figure it is also possible that this specimen has 4 to 5 sacral vert/ribs, since the rib anterior to your sr1 also is distally expanded as is the rib posterior to your sr3. please explain why these are not sr (line 299).

In our judgment, the rib lying immediately posterior to ‘sr3’ on the left side of HFUT YZSB-19-109 does not have an expanded distal end and closely matches the first caudal rib of *Largocephalosaurus* in morphology (compare with Li et al. 2014 [GeolMag]:Figures4i, 5d and 6a), so we accordingly identify it as a caudal rib. However, we agree with Reviewer 2 that the exact number of sacral ribs/vertebrae is difficult to establish in the new taxon, because the posterior dorsal/sacral ribs are partially obscured by the overlying ilium and ischium. The difficulties of establishing the correct number of sacral ribs were added to the manuscript (lines 315–320):

”Four complete posterior dorsal/sacral vertebrae are preserved and exposed in left ventrolateral view. Two sacral vertebrae can be identified by the presence of associated sacral ribs with clearly expanded distal ends. However, the distal ends of the first two ribs lying immediately posterior to the last unambiguous dorsal rib are obscured by the overlying ilium and ischium, so it is not possible to confidently determine whether they represent the last two dorsal ribs or the first two sacral ribs (Figure 6A).”

All of the parts of the manuscript that mentioned the number of sacral vertebrae/ribs were updated in accordance with this anatomical revision.

5. [a rugose dorsolateral surrface] please show/add in figure (line 310).

A label for the rugose dorsal surface of the neural arch (‘rgs’) was added to Figure 6B.

6. aff Eusaurosphargis from Winterswijk does have uncinate processes on dorsal ribs but maybe this depends on anatomical position and ontogeny ? (line 333)

It IS stated here that the dorsal ribs of *Eusaurosphargis* DO possess uncinate processes and thus differ in this respect from the ribs of HFUT YZSB-19-109, in which uncinate processes are absent.

7. [‘rib-basket’] can you please visualize this by a sketch not totally clear what you mean (line 338).

The ‘rib-basket’ is a saurosphargid synapomorphy introduced by Li et al. (2011:309):

“Distally, the [dorsal] rib quickly expands in its anteroposterior dimension to a degree that results in a contact of the successive ribs with one another (Figure 3B). Collectively, the ribs lying parallel to each other form a closed dorsal ‘rib basket.’”

An appropriate explanation was provided alongside the first appearance of the term ‘rib-basket’ in the main text (line 84):

“broadened dorsal ribs forming a closed rib-basket”.

The term was also replaced by ‘armoured’ in the abstract, to improve clarity for the broader readership.

8. As far as I understand the coracoid is an important element to distinguish taxa please add an overview figure (photo or sketch) of the coracoids of all impoartnat taxa (line 413).

A new figure presenting the coracoid morphology of selected Early–Middle Triassic sauropterygomorph taxa (Figure 7 of the revised manuscript) was added to the manuscript and referred to throughout the text.

9. Why ‘Lariosaurus’/in quotation marks (line 419).

As noted by Wang et al. (2022 [iScience]:2–4), *Lariosaurus sanxiaensis* differs from typical representatives of the genus *Lariosaurus* in several anatomical features (most notably in having a sub-oval rather than an anteroposteriorly constricted coracoid) and stratigraphic occurrence (*Lariosaurus* is otherwise known exclusively from the Middle Triassic), raisinig the possibility that *Lariosaurus sanxiaensis* represents a different genus. As a consequence, we used quotation marks for ‘*Lariosaurus*’ *sanxiaensis* in the previous version of the manuscript to indicate its probable taxonomic distinctness. However, neither *Lariosaurus sanxiaensis* nor any Middle Triassic representative of *Lariosaurus* was included in our phylogenetic analysis, and the holotype and referred specimens of *Lariosaurus sanxiaensis* were not studied during the preparation of this manuscript, so we decided to remove the quotation marks in the revised version of manuscript and refrain from discussing the taxonomy of *Lariosaurus sanxiaensis* in the current study.

10. This [a weak connection between dorsal neural arches and centra] is one characteristic of Sauropterygia , summarized in Rieppel 2000; in addition vertebral ossification follows in different groups different patterns, it is thus not necessarily a good indicator for ontogeny (line 549).

We thank the reviewer for pointing this out. Following the reviewer’s suggestion, the information about a weak connection between the neural arch and centrum was removed from the Ontogeny and body size section and included in the description of the vertebrae, where the feature was more appropirately referred to as a sauropterygian synapomorphy. Furthermore, the ontogeny and body size section was removed from the manuscript altogether and the text on body size estimation was moved to the first paragraph of the anatomical description (lines 311–314):

“Articulated anterior and posterior dorsal neural arches preserved without their respective centra indicate a lack of fusion of the neurocentral suture, a paedomorphic feature characteristic for Sauropterygia (Rieppel 2000a) and also present in *Saurosphargis* (Huene 1936) and *Largocephalosaurus* (Li et al. 2014).”

11. You must discuss and compare your [phylogenetic] results with those of Wang et al. 2022 (line 560).

Following the reviewer’s suggestion, the following paragraphs explaining the major differences between the phylogenetic hypotheses of Wang et al. (2022) and the one presented in this manuscript were added to the Phylogeny of Sauropterygomorpha section of the Discussion:

“The results of our phylogenetic analysis differ significantly from the results of a phylogenetic analysis recently published by Wang et al. (2022), in which *Hanosaurus* was recovered as the sister-group to a clade comprising Saurosphargidae + Sauropterygia within a monophyletic Sauropterygiformes. However, we believe that our phylogenetic analysis presents a more accurate topology of sauropterygians and their relatives for the following reasons. Wang et al. (2022) used 16 outgroup (non-sauropterygiform) taxa (15 marine reptiles and a single terrestrial reptile), representing five major reptile lineages, whereas our study included 40 outgroup (non-sauropterygomorph) taxa (19 terrestrial and 21 aquatic reptiles), representing 23 major reptile lineages. The inclusion of only a single terrestrial outgroup taxon – *Youngina* – in the analysis of Wang et al. (2022) is problematic because *Youngina* likely represents a taxon rather distantly related to the Mesozoic marine reptile clade which includes Sauropterygia (Figure 8; see also Simões et al. 2022). Distantly related taxa likely share fewer character states with derived taxa (homoplasy accumulation through time), so a phylogenetic analysis containing a single, distantly related outgroup taxon will likely fail to adequately capture important character transformation sequences, in contrast to a phylogenetic analysis in which outgroups are more comprehensively sampled (Wilberg 2015). Furthermore, the phylogenetic analysis of Wang et al. (2022) used 181 morphological characters, in contrast to 221 characters included in our study. Greater character sampling has been demonstrated as an important factor increasing the accuracy of phylogenetic reconstructions (Wiens 2006), which also favours the results obtained by our analysis.

The different topologies recovered by both studies are likely also partially a consequence of differences in character scoring. For example, Wang et al. (2022) scored the humerus of *Hanosaurus* as ‘rather straight’ (plesiomorphic state), similar to the humerus of *Youngina* and other non-sauropterygiform marine reptiles. However, in our opinion, the humerus morphology of *Hanosaurus* matches the typical ‘curved’ morphology (derived state) characteristic of saurosphargids, placodonts and the majority of eosauropterygians (Rieppel 2000a). Wang et al. (2022) also scored *Hanosaurus* into an updated version of the phylogenetic matrix of Neenan et al. (2013). *Hanosaurus* was recovered as the earliest-diverging sauropterygiform in this analysis as well, but its humerus was also scored as plesiomorphic (‘rather straight’) in the character-taxon matrix. Interestingly, this updated analysis of Neenan et al. (2013) included a more comprehensive outgroup (non-sauropterygiform) sample (12 terrestrial and 5 aquatic taxa, representing 12 major reptile lineages) than the analysis of Wang et al. (2022) and recovered *Eusaurosphargis* and *Helveticosaurus* as successive sister-groups to Sauropterygia – a result similar to the one obtained in this study (Figure 9).”

We also referred to another recently published paper on the phylogenetic interrelationships among reptiles (Simões et al. 2022 [SciAdv]) throughout the text.

12. ploytomy (line 563).

Corrected to ‘polytomy’.

13. Did you also include in another version Cyamodontidae/heavily armored placodonts ? I am wondering if this would make any change (line 570).

The phylogenetic matrix used in this study was originally constructed to include early-diverging representatives of various reptile groups. However, the morphology of cyamodontoid placodonts is very derived in comparison with their early-diverging members (modified skull, dentition, girdles, body armour). Some of these derived features (such as the anteroposteriorly short premaxilla) are convergent with similar features seen in early-diverging sauropterygomorphs (anteroposteriorly short premaxilla in *Eusaurosphargis*), which might bias the results of the phylogenetic analysis presented in this manuscript if cyamodontoid placodonts were included. To prevent some of these convergent character states from influencing the results of our analyses, the inclusion of cyamodontoid placodonts into our phylogenetic matrix would also necessitate the addition/revision of several characters and character states, which is beyond the scope of the present study. We therefore do not include cyamodontoid placodonts in the current study, and consider them as derived taxa of limited relevance for addressing our specific research questions.

14. "small" etc is a difficult character which should be defined (for example give a ratio or smaller than XY) (line 604).

A statement explaining that in this context ‘small’ means anteroposteriorly short, together with an appropriate reference containing definitions of both character states used to describe the relative size of the premaxilla (Chen et al. 2014 PloS One), were added to the revised version of the manuscript (lines 674–678).

15. Sauopterygomorpha (line 667).

Corrected to ‘Sauropterygomorpha’.

16. Please add reference for this statement (line 676).

Appropriate reference is already cited at the end of the sentence:

“Both of these character states are considered as typical for terrestrial taxa and contrast with the enlarged premaxillae and posteriory displaced external narial openings characteristic for marine reptiles (Chen et al. 2014a).”

17. posteriory (line 677).

Corrected into ‘posteriorly’.

18. ? to my knowledge there is no bone histological study on Eusaurosphargis so far; microanatomy based on Ct data are mentioned in Scheyer et al. 2017 (line 679).

Replaced ‘histology’ with ‘microanatomy’.

19. Unclear statement please rephrase to be clearer (line 704).

Replaced ‘placodonts’ with ‘placodont fossils’ for clarity.

20. Please add results and discuss those from Wang et al. 2022 which have a completely different hypo (line 748).

Please see response to comment #11.

21. demonstreates (line 753).

Corrected into ‘demonstrates’.

22. eosauropterygia (line 767).

Sentence containing typo was deleted as it contained a factual error.

23. unclear/why (lines 777–778; 792–793).

Lines 777–778 and 792–793 were edited in order to more clearly explain that the dermal armour present in early sauropterygian ancestors most likely countercated buoyanacy and enabled these animals to explore the bottom of the shallow sea in search of food:

“Our phylogenetic analysis indicates the important role of body armour in sauropterygian evolution (Figure 10). Dermal armour was likely an important preadaptation that allowed colonisation of the shallow marine realm by a *Eusaurosphargis*-like ancestor, enabling it to counteract buoyancy and walk on the bottom of the shallow sea in search of food (Houssaye 2009). Elaboration of dermal armour occurred in the shallow marine placodonts and saurosphargids, perhaps as a response to predation pressure (Liu et al. 2014; Chen et al. 2014b; Qiao et al. 2020). The reduction and complete loss of dermal armour then occurred in the lineage leading to Eosauropterygia, and was likely involved with the evolution of active predation, an efficient swimming style and increasing adaptation to a pelagic lifestyle.”

“These evolutionary parallels seem to demonstrate that dermal body armour could have been a possible prerequsite (preadaptation) for the invasion of the shallow marine realm in different diapsid clades, which allowed for buoyancy reduction and exploration of the shallow marine environment in search of food.”

24. Please shortly discuss some of the characters which lead to this result (line 799).

The synapomorphies of Archelosauria have already been listed in the Phylogenetic Results section and the following brief discussion was added to the end of the final paragraph of the Discussion section:

“Archelosauria are supported in our analysis by four unambiguous synapomorphies associated with the morphology of the skull and shoulder girdle (see above). This demonstrates that innovations of the cranium linked with the evolution of sensory organs and the feeding apparatus, as well as changes in locomotion, perhaps underlie the evolutionary success of archelosaurian reptiles.”